# Single-cell transcriptomic analysis of mouse neocortical development

Lipin Loo[1], Jeremy M. Simon [1,2,3], Lei Xing[1], Eric S. McCoy[1], Jesse K. Niehaus[1], Jiami Guo[1,2], E.S. Anton[1,2] & Mark J. Zylka[1,2]

The development of the mammalian cerebral cortex depends on careful orchestration of proliferation, maturation, and migration events, ultimately giving rise to a wide variety of neuronal and non-neuronal cell types. To better understand cellular and molecular processes that unfold during late corticogenesis, we perform single-cell RNA-seq on the mouse cerebral cortex at a progenitor driven phase (embryonic day 14.5) and at birth—after neurons from all six cortical layers are born. We identify numerous classes of neurons, progenitors, and glia, their proliferative, migratory, and activation states, and their relatedness within and across age. Using the cell-type-specific expression patterns of genes mutated in neurological and psychiatric diseases, we identify putative disease subtypes that associate with clinical phenotypes. Our study reveals the cellular template of a complex neurodevelopmental process, and provides a window into the cellular origins of brain diseases.

[1] Department of Cell Biology and Physiology, UNC Neuroscience Center, The University of North Carolina at Chapel Hill, 115 Mason Farm Road, Chapel Hill, NC 27599, USA. [2] Carolina Institute for Developmental Disabilities, The University of North Carolina at Chapel Hill, Campus Box #7255, Chapel Hill, NC 27599, USA. [3] Department of Genetics, The University of North Carolina at Chapel Hill, Campus Box #7264, Chapel Hill, NC 27599, USA. These authors contributed equally: Lipin Loo, Jeremy M. Simon. Correspondence and requests for materials should be addressed to M.J.Z. (email: zylka@med.unc.edu)

The mammalian cerebral cortex develops via a complex sequence of cell proliferation, differentiation, and migration events. In the mouse, cortical progenitors rapidly divide between embryonic day 11.5 (E11.5) and birth (P0), giving rise to six neocortical layers[1]. Neural stem cells in the ventricular zone (VZ), intermediate progenitors of the subventricular zone (SVZ), and radial glia (RG) in the cerebral cortex undergo a series of symmetric or asymmetric divisions to produce more intermediate progenitors or pyramidal neurons[2]. Terminally differentiated neurons migrate radially to their final destination, forming cortical lamina in an inside-out manner. Dynamic expression of transcription factors such as COUP-TF-interacting protein 2 (CTIP2; also known as BCL11B), zinc-finger transcription factor FEZF2 and special AT-rich sequence binding protein 2 (SATB2), tightly regulate this laminating process and confer specific axonal projection characteristics to subcerebral (SCPN), corticothalamic (CThPN), and callosal projection neurons (CPN), while diffusible factors such as FGF8 and WNT control the relative size and position of cortical areas[1].

During this time, GABAergic interneurons differentiate from progenitor cells in the VZs of subpallial ganglionic eminences and migrate tangentially into the cortex. Instead of extending a single leading process in the direction of migration, interneurons can extend multiple processes to adjust their polarity in response to chemotactic cues and eventually populate all layers of the cortex[3,4]. The final cortical location of interneurons is defined by expression of genes such as *Dlx1/2*, *Nkx2.1,* and *Lhx6*[5]. This process gives rise to cardinal classes of parvalbumin (PV), somatostatin (SST) and vasointestinal peptide (VIP) expressing interneurons.

Genetic and environmental factors that perturb processes described above can impair intellect and increase the risk for neurodevelopmental disorders such as autism spectrum disorder (ASD)[6]. It is postulated that common convergent pathways are affected in neurodevelopmental disorders, resulting in improper lamination, expansion or reduction in certain cortical layers, and excitatory-inhibitory imbalances[6]. While mice are routinely used to study neurodevelopmental processes and to model brain disorders, there is currently no comprehensive catalog of cells that make up the normally developing mouse cerebral cortex.

Transcriptomics have deepened our understanding of the genetic programming underlying cortical development in various species[7–9]. Unlike bulk transcriptomics, which interrogates average gene expression in a heterogeneous tissue, single-cell transcriptomics can be used to profile gene expression in individual cells and uniquely classify neural cell types based on combinatorial gene expression[10]. The recent advent of massively parallel high-throughput droplet-based profiling techniques have further hastened the adaptation of single-cell RNA-seq in cataloging cells in the central and peripheral nervous system of adults[11–15].

Here, we use Drop-seq to characterize the cellular composition of the developing mouse cortex at two key times in development —embryonic day 14.5 (E14.5), representing a progenitor-driven stage, and birth (P0), when neurons corresponding to all six cortical layers have been born and gliogenesis has begun. We identify distinct cortical layer-specific cell types, which express the longest genes[16], multiple progenitor-like cell types, including *Eomes*[+] (*Tbr2*[+]) progenitors, GABAergic interneurons, and non-neuronal cells, such as endothelial cells and microglia.

## Results

### Single-cell transcriptomics of the E14.5 and P0 cortex. We transcriptionally profiled a total of 18,545 mouse neocortical cells at two key times of corticogenesis using Drop-seq: 10,931 cells at

embryonic day 14.5 (E14.5) from six biological replicates, and 7614 cells at birth (P0) from three biological replicates (Fig. 1a, Supplementary Table 1)[11,17]. Single-cell libraries were sequenced to a median depth of ~12,000 reads/cell, detected a median of ~2500 transcripts per cell, and represented a median of ~1600 of genes per cell (Fig. 1a, Supplementary Figures 1, 2). This depth is on par with or exceeds similar studies[11,12,14]. To increase the precision of unbiased clustering, we developed an iterative cell type refinement method (Fig. 1a, Supplementary Figure 3, see Methods) and identified 22 principal cell types at each age (Fig. 1b, c, Supplementary Data 1). Using t-Distributed Stochastic Neighbor Embedding (t-SNE[18]) overlaid with expression levels of the broad cell-type markers *Neurod6* (excitatory neurons), *Gad2* (inhibitory neurons), *Eomes* (*Tbr2*, neuronal progenitor), and *Mki67* (proliferating and glial), we observed separation of these broad cell-type markers and their constituent cell types (Fig. 1b–e).

**Characterization and validation of cortical cell types**. To assign biological labels to each of these cell types, we first identified cluster-specific marker genes, similar to other single-cell transcriptomic studies[11,12] (Fig. 1b, c, Fig. 2, Supplementary Figure 4). Each cell type exhibited similar overall transcript levels and cell proportions among biological replicates, suggesting that none of the clusters were skewed by residual batch effects (Supplementary Figure 2 and 5). For each identified marker gene, we next validated that those genes were expressed in the correct cell types, in the correct cortical regions/layers, and at the correct age using in situ hybridization data (Eurexpress, Allen Institute of Brain Science, GENSAT) (Supplementary Figure 6–13). We assembled these annotations, along with additional references confirming the identity of these cell types and their marker genes, as well as pathway-level enrichment analyses that describe the predominant transcriptional signatures of each cell type in Supplementary Data 2.

We identified Layer I (Cluster 17-E and 19-P) cells at both time points, which expressed canonical Cajal-Retzius cell markers *Reln*, *Trp73*, *Lhx1*, and *Lhx5* (Supplementary Figures 4, 6, and 10, Supplementary Data 2). Five excitatory neuron clusters were also present at both time points. Lower-layer neurons were present at E14.5 and were similar to their P0 counterparts, as expected given the timing of cortical layer formation[17]. All E14.5 excitatory neuron clusters (5-E, 13-E, 3-E, 7-E, and 2-E) broadly expressed *Bcl11b*, a deep layer marker (Supplementary Figures 4 and 6, Supplementary Data 2). Layer V–VI neurons could be further distinguished by genes characteristic of their function or that demonstrated regional specificity. For example, Clusters 5-E and 13-E both expressed *Fezf2*, which is normally expressed at high levels in Layer V SCPN and at lower levels in Layer VI CThPN[19]. These two clusters could be further segregated spatially by expression of *Crym*, which is expressed more caudally[20], and *Mc4r*, which is expressed more rostrally[20] (Supplementary Figures 4, 6, and 14 and Supplementary Data 2). Cluster 7-E also showed regional specificity, given its expression of *Tfap2d*, which is expressed more rostrally[21] (Supplementary Figures 4 and 6). We also identified three classes of Layer II–IV (upper-layer) neurons in the P0, but not E14.5, cortex, consistent with the later birthdate of upper-layer neurons. Each of these clusters (Clusters 1-P, 4-P, 15-P) expressed *Satb2* and *Pou3f1*, and 4-P was further specified by expression of *Nrgn*, *Inhba*, and *Pvrl3* (Supplementary Figure 10).

Newly generated interneurons migrate tangentially from the ganglionic eminences and populate all layers of the cerebral cortex[3]. We identified two interneuron types, Int1 (Clusters 1-E and 5-P) and Int2 (Clusters 12-E and 14-P), that were present at

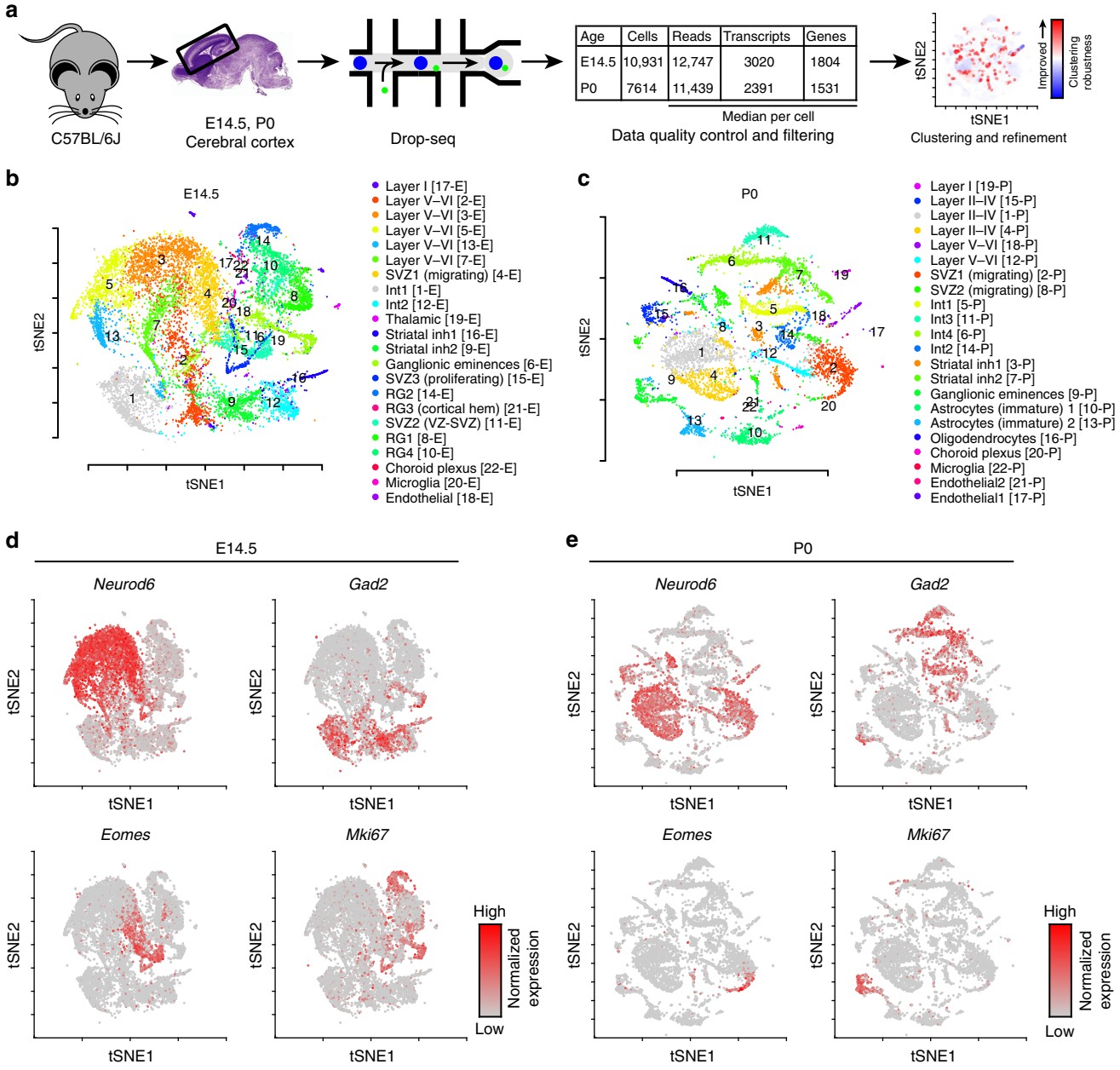

**Fig. 1** Overview of the experimental approach and cell cluster analyses. **a** Cortical cells were isolated from E14.5 and P0 C57BL/6J mice across multiple biological replicates (*n* = 6, E14.5; *n* = 3, P0). A novel iterative clustering framework was used to identify and refine cell types, which improved the cluster robustness of most cells. **b** t-SNE visualization of the 22 clusters identified in the E14.5 cortex. **c** t-SNE visualization of the 22 clusters identified in the P0 cortex. **d** Visualization of E14.5 cells and **e** P0 cells overlaid with gene expression information of canonical marker genes, *Neurod6* (excitatory neuron), *Gad2* (interneuron), *Eomes* (*Tbr2*, neuronal progenitors), and *Mki67* (proliferating and glia). The expression is depicted from gray (low) to red (high)

E14.5 and P0 and expressed high levels of *Lhx6*, a transcription factor associated with PV and SST interneurons[22] (Supplementary Figures 4, 7, and 11). *Sst*, but not *Pv*, was detected in Int1 and Int2 at these ages, as expected[23]. We also identified two interneuron classes unique to the P0 cortex, namely Int3 and Int4. Int3 (Cluster 11-P) expressed canonical markers of vasoactive intestinal peptide (VIP) cells, including *Htr3a*, *Npas1*, and *Adarb2*[22]. Int4 (Cluster 6-P) expressed *Cdca7*, a marker of some SST⁺ and PV⁺ interneurons[22], but did not express *Sst* at this stage. Int4 expressed high levels of *Sp9*, *Tiam2*, and *Dlx5*— general transcriptional and migratory markers[24,25]—suggesting that Int4 is an immature/migrating interneuron cluster (Supplementary Figure 11).

We also identified three SVZ clusters—SVZ1, SVZ2, and SVZ3 (Clusters 4-E, 11-E, and 15-E)—that were marked by strong expression of *Eomes* (*Tbr2*)[26] at E14.5 (Supplementary Figures 4 and 8). Two similar clusters were identified at P0—Clusters 2-P and 8-P (SVZ1 and SVZ2)—that, in addition to *Eomes*, expressed markers for newborn and migrating neurons (*Sema3c* and *Neurod1*)[24,27] (Supplementary Figures 4 and 12). Cells in Cluster 8-P expressed SVZ markers as well as markers of both excitatory (*Neurod6*) and inhibitory (*Calb2*) neurons. To confirm that this cell type exists, we performed immunofluorescence staining in P0 *Neurod6:CRE* mice; the expression of CRE faithfully recapitulates endogenous *Neurod6* promoter activity[28]. We observed a migratory stream of CRE⁺CALB2⁺ cells in the corpus callosum

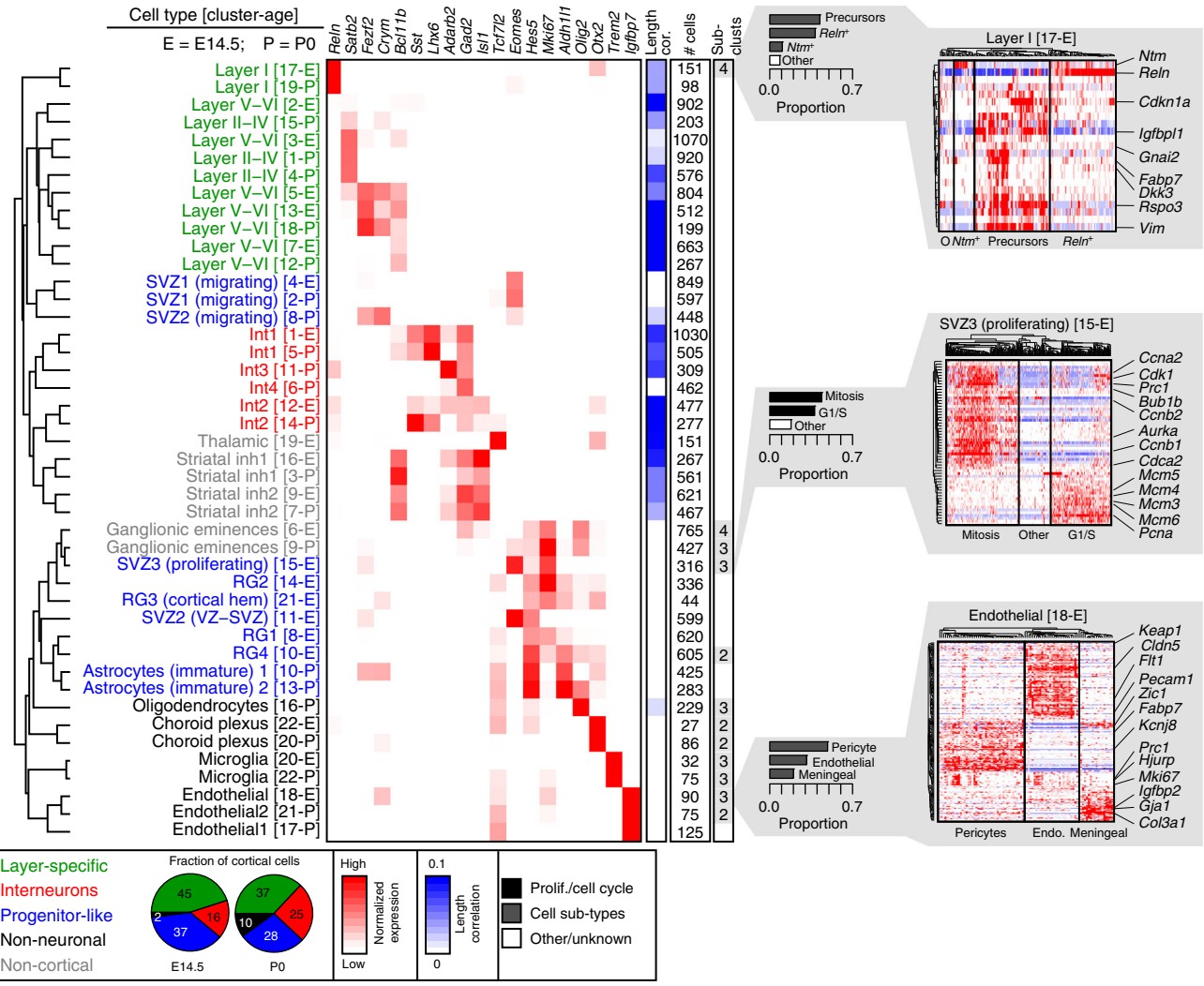

**Fig. 2** Characterization of cell types in the developing cortex. Cell types were grouped into categories (colored), based on their functional identity and transcriptional similarity (Pearson correlation distances, dendrogram). Correlation of expression with gene length provided on a scale of white to blue. Total number of cells identified for each cluster is provided. Fractional proportions of cortical cells, averaged across all biological replicates, is depicted as a pie chart; non-cortical cells were excluded. Number of cellular sub-clusters for each cell type is indicated, as well as three sub-cluster examples. All sub-clusters are fully characterized in Supplementary Materials

as well as some similarly labeled cells in the cortical plate (Supplementary Figure 15). These cells exhibit similar localization and expression patterns to the Rostral Migratory Stream, a population of cells that migrate to the cortex and olfactory bulb postnatally[29,30].

The four RG clusters (8-E, 13-E, 21-E, and 10-E) at E14.5 expressed RG markers *Hes1*, *Hes5*, *Pax6*, and *Ednrb*, and proliferation markers *Mki67* and *Top2a* (Supplementary Figures 4 and 8, Supplementary Data 2). At P0, *Hes5* was expressed in two immature astrocyte cell clusters (10-P and 13-P), which also expressed mature astrocyte markers—*Aqp4* and *Aldh1l1*[31] (Supplementary Figures 4 and 12). These two mature astrocyte markers were undetected in any cell type at E14.5.

We detected oligodendrocytes (Cluster 16-P), marked by *Olig2* expression, in the P0 cortex only (Supplementary Figures 4, 9, and 13). Non-neuronal cells such as choroid plexus (22-E and 20-P, *Otx2+*), microglia (20-E and 22-P, *Trem2+*), and endothelial cells (18-E, 17-P and 21-P, *Igfbp7+*)[32] were also detected. We also identified other cell types from adjacent non-cortical tissues— ganglionic eminences (*Gad2+*, *Mki67+*), striatal inhibitory neurons (*Isl1+*, *Gpr88+*, *Rxrg+*) and thalamus (*Tcf7l2+*, *Syt13+*)

(Supplementary Figures 4, 7, 8, 11, and 12, Supplementary Data 2).

**Transcriptional similarity of cell types across ages and species.** To assess the overall proportions of cellular classes at each age, we first grouped the cortical cells into four broad categories (layer-specific, interneurons, progenitor-like, and non-neuronal) (Fig. 2). Nearly 40% of cells at E14.5 were progenitor-like, including multiple classes of RG and intermediate progenitors localized to the VZ and sub-ventricular zone, but represented only 28% of cells by P0, as expected[33]. The ratio of excitatory to inhibitory neurons was in line with previous estimates that range from 2:1 to 5:1[34,35]. Additionally, the P0 cerebral cortex contained a greater proportion of non-neuronal cells relative to the E14.5 cortex, which is consistent with the known timing of glial proliferation[33].

Next, we assessed the specific cellular composition as well as the transcriptional similarity of these identified cell types across ages using hierarchical clustering with correlation-based distances (Fig. 2). These relative correlations permit the side-by-side

comparison of cell types within and across ages. Many of the identified cell types were more similar to one another at E14.5 and P0 ages than they were to other related cell types. For example, Layer I cells of the E14.5 cortex correlated best to that of the P0 cortex, and similar correlated pairs were observed for Int1 and Int2 interneurons as well as several glial cell types. We also observed novel correlations that, when combined with the underlying expression patterns, are suggestive of early fate specification. For example, Cluster 3-E expressed an upper-layer CPN marker (*Satb2*), a lower-layer marker (*Bcl11b*), a migratory marker *Tiam2*, and *Pou3f1*, a transcription factor that is expressed in Layer II–III neurons during their migration and differentiation (Supplementary Figures 4, 6, and 10)[17,36]. This cluster was most similar to an upper-layer CPN at P0 (Layer II–IV; Cluster 1-P). These cells may therefore be destined to become upper-layer CPN, some of which are known to be born around E14.5[37].

We next sought to identify relationships between the cell types we identified to previously described excitatory neuron populations in the developing cortex[7] and to single cell studies of the adult cortex[22,35]. Using correlation analyses (Supplementary Figure 16a), we found that E14.5 deep-layer pyramidal clusters (Clusters 3-E, 5-E, 7-E, 13-E) and P0 upper and deeper layer pyramidal neurons clusters (Clusters 1-P, 4-P, 12-P, 18-P) correlated with the developing pyramidal neuron populations from all ages sampled by Molyneaux et al.[7] P0 Layer V–VI neurons (Cluster 18-P) correlated with S1PyrL5 (Zeisel[35]) and with multiple clusters of Layer VI neurons (Tasic L6b-Serpinb11, L6b-Rgs12, L6a-Sla, and L6a-Mgp[22]). We observed similar correlations for certain classes of interneurons, in addition to microglia, endothelial cells, oligodendrocytes (only with oligo-dendrocyte precursor cells, Tasic OPCs[10]), choroid plexus, and astrocytes.

Human cortical development and composition are very similar to that of the mouse, though primates have an additional form of RG known as outer radial glia (oRG)[38]. We performed a correlation analysis with a recently published catalog of human fetal cortical cell types[39]. We focused on a range of 7–11.5 post-conception weeks (pcw) and 20–23 pcw, as these human ages are analogous to E14.5 and P0 mouse ages, respectively[40,41]. In the younger cortex, we primarily observed concordance among progenitors from multiple RG subclasses, ganglionic eminences, as well as developing excitatory neuron populations and some non-neuronal cell types (endothelial and choroid plexus) (Supplementary Figure 16b). In the older cortex, there was far more correlation between the layer-specific excitatory neuron populations, precursor pools, and glia. Additionally, two classes of interneurons correlated with multiple human interneuron types, including Int1 (*SST/Lhx6*[+]) and Int3 (likely VIP interneurons), indicating that these factors and their transcriptional programs are likely conserved across species. We also observed a correlation between each of our immature astrocytes and human oRG. To examine this correlation further, we looked specifically at the expression of human oRG and other RG markers across all of our cell types (Supplementary Figure 17). We found that markers of human oRG and ventricular radial glia (vRG) were expressed in multiple E14.5 and P0 cell types, suggesting that these cells do not form a cell type that is distinct from other RG in the mouse[38].

**Sub-clustering reveals closely related cell types and states**. To explore the identified cell types further and assess whether any clusters harbored underlying cell sub-types or states, we developed an analytical methodology that focuses on heterogeneously expressed genes within each cluster. We identified sub-clusters in seven of the 22 cell types at E14.5 and in five of the 22 cell types at

P0 (Fig. 2, Supplementary Figures 18, 19). These sub-clusters included: (1) cells in various phases of the cell cycle (RG4 [10-E], SVZ3 (proliferating) [15-E], ganglionic eminences [6-E, 9-P]), (2) highly related but functionally distinct cell types (e.g., pericyte, and meningeal sub-classes of endothelial cells [18-E, 21-P]), and (3) putative activation states (e.g., among microglia [20-E], oli-godendrocytes [16-P], and ganglionic eminences [9-P]). We found no evidence for neuronal activation, as marked by *Fos* and other immediate early genes. Our use of ion channel inhibitors during cell dissociation thus appears to provide a novel way to block dissociation-induced neuronal immediate early gene induction[42].

Interestingly, we also identified four sub-clusters of cells among those identified as Layer I in the E14.5 cortex (Cluster 17-E). In addition to classic mature *Reln*[+] Cajal-Retzius cells, we also detected *Ntm*[+] cells that were mostly *Reln*[-43], as well as a population of precursor-like cells that expressed *Rspo1–3, Dkk3*, and *Vim*. These genes were also highly expressed in RG of the cortical hem (RG3, Cluster 21-E, Supplementary Figure 4), where some Cajal-Retzius cells are known to originate[44,45]. Transcripts of *Rspo1–3* and *Dkk3* could be detected by in situ hybridization in the cortical hem and Layer I as early as E11.5 through mid-embryogenesis[46]. We therefore performed pseudotiming, focus-ing on the gene expression profiles of cortical hem cells (RG3, Cluster 21-E), *Reln*[+] Layer I cells (Cluster 17-E *Reln*[+]), and this precursor-like sub-cluster (Cluster 17-E Precursors) to explore the differentiation trajectory that may link them. We determined that the precursor-like cells of Cluster 17-E represent an intermediate population between immature progenitors of the cortical hem and mature *Reln*[+] Cajal-Retzius cells of Layer I (Fig. 3a). Many of these cells expressed *Sox2*, a marker of multipotent neural stem cells, *Dkk3, Eomes* (*Tbr2*, marker of intermediate progenitors), and some also expressed *Reln* (Fig. 3b). We validated the cellular distribution of DKK3, SOX2, EOMES (TBR2), and RELN using immunofluorescence. We found that DKK3[+] cells were present in both the cortical hem and Layer I (Fig. 3c–h, Supplementary Figure 20). Further, a gradient of DKK3 protein was apparent in the cortical hem, delineating a differentiation trajectory from cortical hem progenitors to mature Cajal-Retzius cells. We identified many cells that expressed a mixture of DKK3, EOMES, and RELN, strongly suggesting a transition between these two states, and corroborating the transcript-level expression patterns described above (Fig. 3a, b). Given that Cajal-Retzius cells can originate from the cortical hem[44,45] our data suggest that the Cluster 17-E precursor-like sub-cluster of cells originated from the cortical hem and are destined to give rise to mature Cajal-Retzius cells of Layer I.

**Putative disease subtypes based on expression profiles**. Cortical dysfunction is implicated in neurological and neuropsychiatric diseases, including amyotrophic lateral sclerosis (ALS), Alzhei-mer's disease (ALZ), ASD, ciliopathies (CIL), and schizophrenia (SCZ). Genes that increase the risk for these diseases were recently identified[47–52]. To determine if these disease-associated genes are expressed broadly or specifically in developing cortical cell types, we hierarchically clustered the cellular expression profiles of these genes. We defined classes of disease-linked genes with shared cellular expression at the individual cell type level and summarized across broader cellular classes (Figs. 4, 5, Supple-mentary Figures 21–24, see Methods).

Genes mutated in ciliopathies segregated into four subtypes (Fig. 4). The largest class of genes exhibited expression almost exclusively in the choroid plexus (Subtype 1), whereas the other subtypes contained genes expressed primarily in proliferative cells (Subtype 2), neurons (Subtype 3), and glia (Subtype 4). We

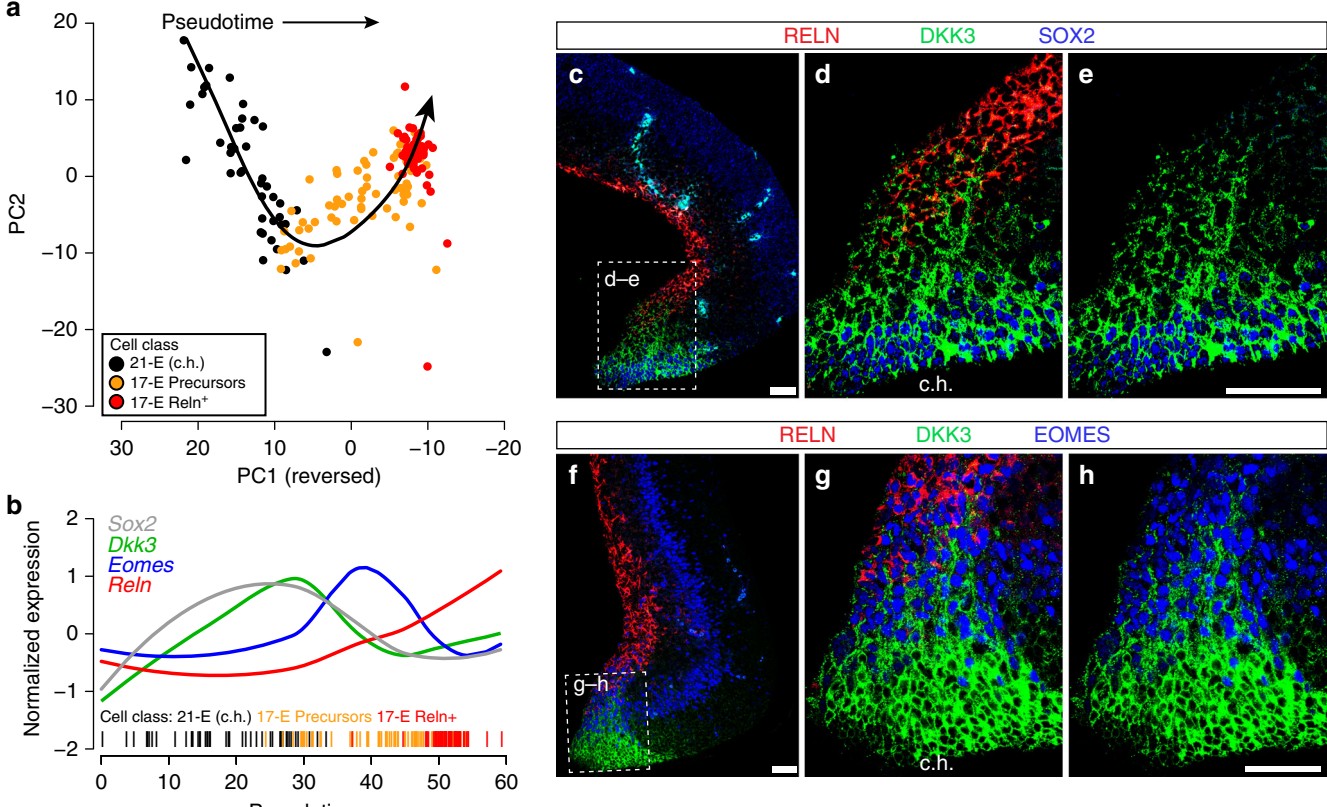

**Fig. 3** Empirical validation of Layer I sub-clusters in the E14.5 cortex. **a** Pseudotiming analysis of cortical hem (c.h.) cells (Cluster 21-E), and *Reln*[+] and precursor-like sub-clusters of Layer I cells. The curvilinear trajectory was fit using Slingshot on a Principal Components Analysis of gene expression Z-scores. **b** Loess-smoothed Z-scored gene expression of four marker genes over pseudotime. **c–e** Co-staining of the cortical hem (boxed) with DKK3 (green), SOX2 (blue), a marker of multipotent neural stem cells, and RELN (red), a marker of differentiated Layer I neurons. Scale bar represents 50 μm. **f–h** Co-staining of the cortical hem with DKK3 (green), EOMES/TBR2 (blue), a maker for intermediate neuronal precursors, and RELN (red). Scale bar represents 50 μm

hypothesized that these subtypes might therefore exhibit differences in the prevalence of certain clinical phenotypes, including microcephaly, hydrocephaly, axonal tract defects, or intellectual disability. We tabulated the number of genes in each expression subtype that were associated with each clinical phenotype, and indeed found a statistically significant association suggesting that these clinical phenotypes were distributed non-randomly among the four subtypes identified (Fisher's exact test, $p = 0.0406$). We then inspected other disorders where genotype-phenotype associations are less well understood (Fig. 5). Ten of the ALS-associated genes (e.g., *C9ORF72*, *Optn*, *Sod1*) were expressed much more highly in non-neuronal cells thereby defining Subtype 2. The majority of the 14 genes linked to Alzheimer's disease (e.g., *Apoe*, *Trem2*, *Picalm*, *Cr1l*, *Cd2ap*) were expressed predominantly in non-neuronal cells, especially microglia. Recent studies suggest that microglia contribute to neurodegeneration in Alzheimer's disease[53,54]. ASD Subtype 1 contained numerous synaptic transmission genes (e.g., *Grin2b*, *Scn2a1*) expressed most highly in neuronal populations, consistent with the previous subtyping based on gene ontology and molecular pathway analyses[55]. The remaining five ASD subtypes contained chromatin modifiers and transcriptional regulators, which gene ontology-based methods generally collapse into a smaller number of groups, but whose expression patterns differed across cell types and age. Genes linked to SCZ segregated into six subtypes, each with a unique pattern of cell- and age-specific expression. Together, these data demonstrate that cell-specific expression patterns but also age (and likely other factors,

including sex) are important in determining the effects of these mutations on the brain and cellular vulnerabilities associated with expressing disease-linked gene mutations.

**Web-based tools for data visualization and exploration**. Lastly, we created a web-based tool for exploration and visualization of our data from each age, accessible from http://zylkalab.org/data. Users can search for individual genes and explore cell type and age-specific expression patterns (Fig. 6).

## Discussion
We assembled a catalog of cell types in the developing mouse cerebral cortex, described the spatial and temporal expression patterns of hallmark genes, and uncovered underlying cellular states and sub-clusters using a novel iterative analytical framework. Our study enables the direct molecular comparison of all cell types of the developing cortex under the same experimental scheme. Further, our identification of cellular sub-clusters as well as proliferative, migratory, and activation states demonstrates not only the power of single-cell transcriptomics, but also the ability to leverage intra-cluster heterogeneity to extract new or additional information about cell states. Indeed, our sub-clustering analysis identified a small class of cells that linked mature Cajal-Retzius cells in Layer I to their immature progenitors, which we were able to confirm via in silico pseudotiming and immunofluorescence staining. This does, however, complicate the emerging discussion of what constitutes a "cell type" or "cell state," and underscores

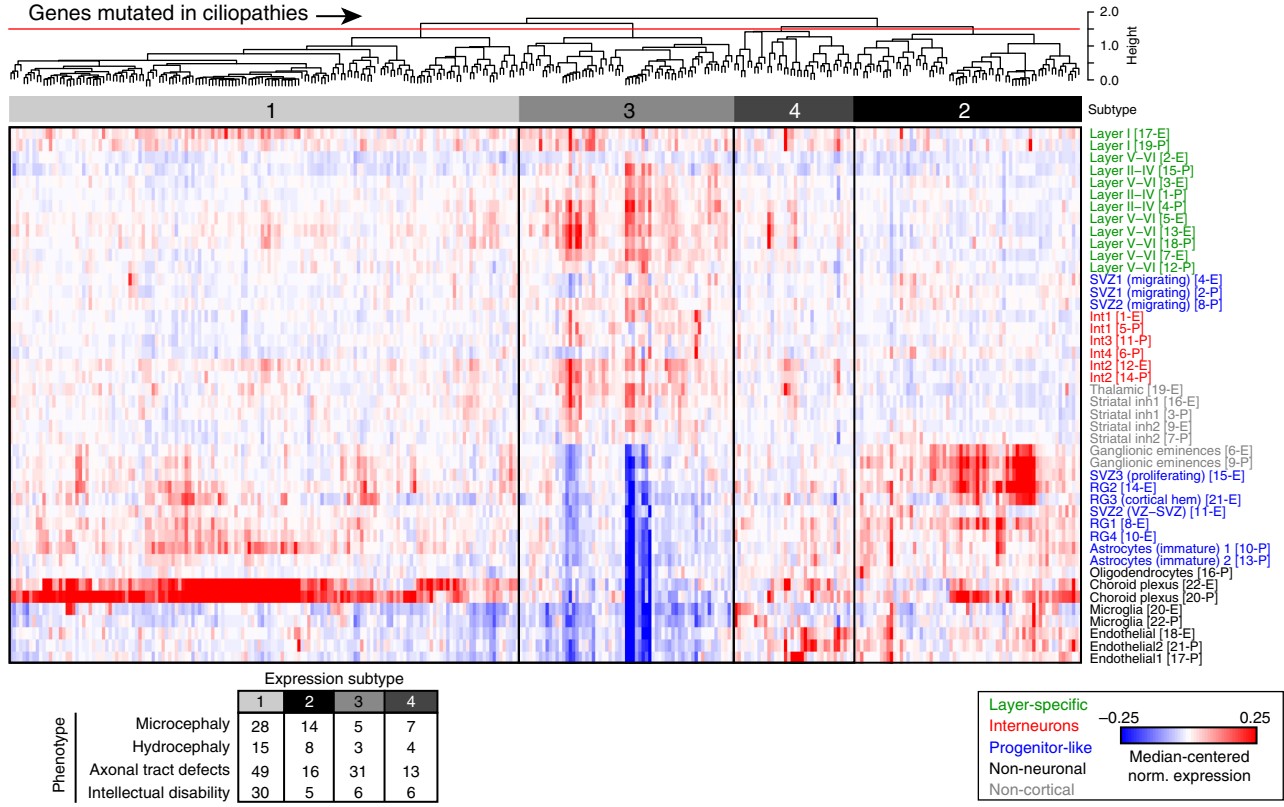

**Fig. 4** Putative ciliopathy disease subtypes associate with clinical phenotypes. Normalized expression values for genes mutated in ciliopathies were hierarchically clustered, and the gene dendrogram (Pearson correlation) was cut at a set height (red line) to identify clusters of genes with similar cell-type- and age-specific expression profiles. Counts indicating how many genes within a given subtype associated with a specific clinical phenotype are provided as a table. Median-centered normalized expression values are depicted from blue (−0.25) to white (0.0) to red (0.25)

how scientific technology has surpassed what we currently can describe semantically. Cellular categorization is often hierarchical in nature, such as in the fully developed adult brain[22,35]. However, cellular differentiation occurs along a continuum rather than in discrete steps, causing cell nomenclature to blur, as we found in transitioning and immature cell types.

We also performed extensive gene expression correlation analyses between the cell types we identified in the developing cortex and cell types identified in the fetal and adult brains of mice and humans. We identified expected parallels and some unexpected discrepancies. For example, we did not observe significant overlap among all interneuron populations across developmental time. Whether this is attributed to differences in how interneurons change their transcriptional program over their developmental trajectory relative to other cell types, or disparity in the number of interneurons sampled would require further study. It is well known that many of the commonly used interneuron markers are induced by neuronal activity and are generally not detectable at the protein level in newborn mice[23]. These analyses also suggest that some, but not all of the cellular complexity of the adult brain, particularly among neurons, is established early in development. Emerging methodologies to simultaneously trace cellular lineages and changes in gene expression over developmental time[56,57] might be best suited to study when these neurons are born, how and where they proliferate and migrate, and where they ultimately reside.

Our data also suggest that disease-linked gene mutations might form robust groupings based on their cell-type- and age-specific expression profiles. While single-cell transcriptomic studies have identified cell types that are affected in Alzheimer's disease or

conditions such as food deprivation[54,58], whether these putative disease subtypes exhibit differences in patient phenotypes is an important question that warrants further exploration. Our data on genes mutated in ciliopathies suggest that expression-based genetic subtyping may have prognostic power. The pathogenesis of neurodevelopmental disorders such as ASD is thought to initiate during early to mid-fetal brain development[59,60], corresponding to ~E14.5 in mice[40,41], therefore understanding how cell types in this developmental window differ in their vulnerability to mutation may be crucial to determining disease etiology. Other diseases and disorders with a less clear link to neurodevelopmental defects, such as Alzheimer's disease and schizophrenia, may require integration with cell-type-specific gene expression data from fetal to aged brains in order to better link genotypes and phenotypes. Other factors, including sex and individual- or population-level genetic differences, will likely also influence the cellular environment within which these mutations manifest. In conclusion, our single-cell data provide an essential resource for future studies directed at understanding how genetic and environmental factors affect cell composition, cell states, and cell fates during early mouse brain development.

## Methods

**Mouse handling and timed matings**. All procedures used in this study were approved by the Institutional Animal Care and Use Committee at the University of North Carolina at Chapel Hill. Mice were maintained on a 12 h:12 h light:dark cycle and given food and water ad libitum. Timed matings were set up in the evening, using one male and two female C57BL/6J mice (Jackson Labs) per breeding cage. The male mouse was separated from female mice the next morning. 14 days after separation, pregnant female mice were euthanized and embryonic day

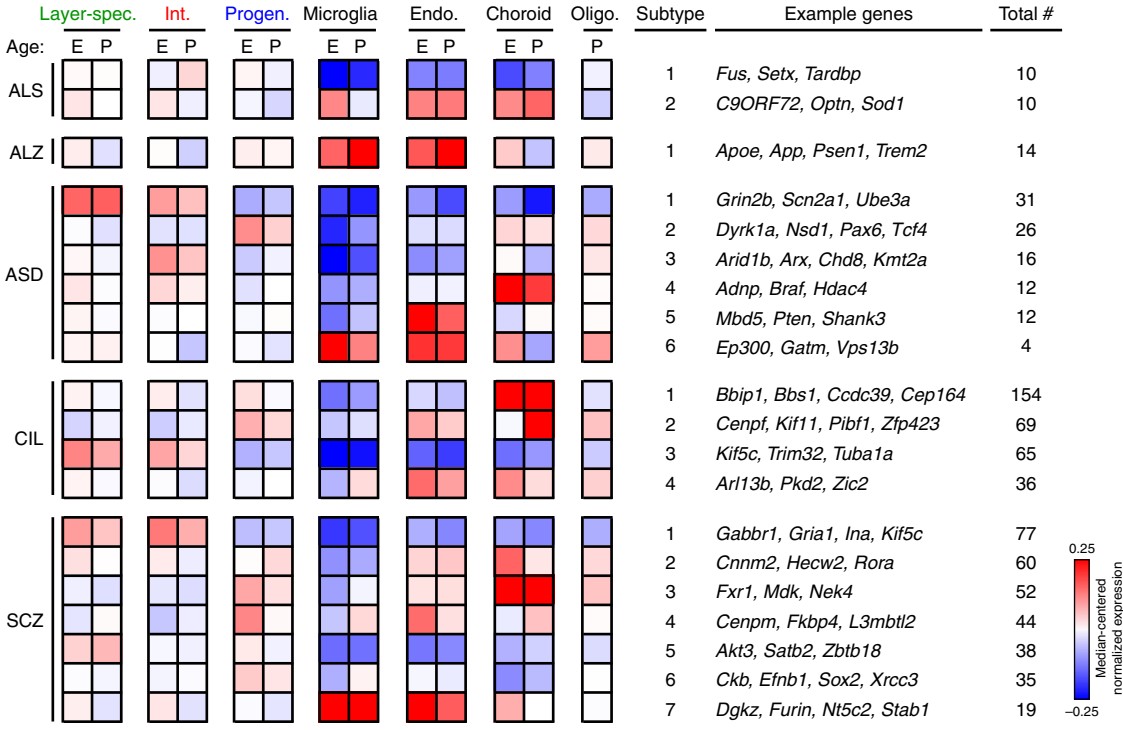

**Fig. 5** Identification of putative disease subtypes using hierarchical clustering. Normalized expression values for genes linked to ALS, Alzheimer's disease (ALZ), autism spectrum disorders (ASD), ciliopathies (CIL), and schizophrenia (SCZ) are summarized by collapsing cell types (using median) into their broader categories (colored); non-neuronal types were not collapsed. Full datasets showing expression of each gene in each cell type is provided in Supplementary Materials. Median-centered normalized expression values are depicted from blue (−0.25) to white (0.0) to red (0.25)

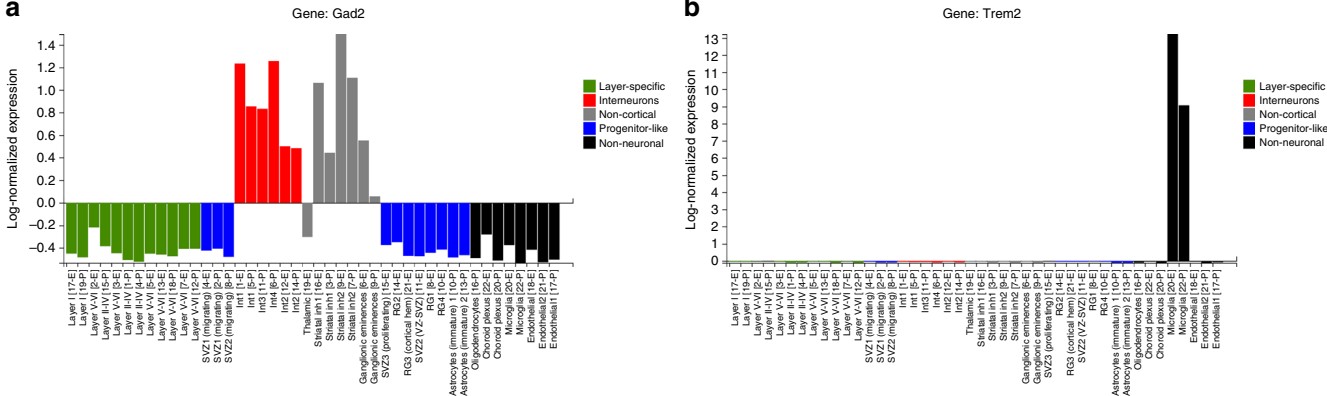

**Fig. 6** Snapshots of the web-based data visualization tool. **a**, **b**. Normalized expression (mean of all cells within cluster) of query genes illustrated as a live and exportable barplot

14.5 (E14.5) embryos were collected for dissections. For P0 pups, cages were monitored daily from E18.5 to postnatal day 0 (P0) for newborn pups.

**Cortical dissections and single cell suspension preparation**. Cerebral cortices (both halves) from E14.5 and P0 mice were dissected in neurobasal medium and rinsed with Hank's Balanced Salt Solution (HBSS; 14175095 Gibco)[61]. All procedures were done at room temperature, unless otherwise stated. Cortices were then incubated for 30 min at 37 °C in papain (1 vial diluted with 2.5 mL of HBSS; Pierce 88285) with DNase I (20 mg mL$^{-1}$; D4513 Sigma) in Ca$^{2+}$ and Mg$^{2+}$ free HBSS. Next, 1 mL of neurobasal medium containing 5% FBS was added to the cortical mixture and triturated to deactivate the papain. The cells were centrifuged for 2 min at 4600 × $g$, washed twice, and resuspended in Ca$^{2+}$ and Mg$^{2+}$ free HBSS with ion channel inhibitors (5 μM TTX ab120054 Abcam, 25 μM DL-AP5 ab120004 Abcam, 5 μM DNQX 2312 Tocris). Cells were kept on ice until the microfluidics run. A total of nine replicates were prepared from two developmental time points (6 replicates for E14.5 embryos and 3 replicates for P0 pups). Each replicate contained cortical cells from male and female littermates.

**Drop-seq procedure**. Drop-seq was performed largely as described[11]. Briefly, cortical cells were diluted to an estimated concentration of 400 cells μL$^{-1}$ in Ca$^{2+}$ and Mg$^{2+}$ free HBSS with ion channel inhibitors and HEK293T cells were spiked in at a concentration of 3% of total cells (12 cells μL$^{-1}$) while barcoded beads (ChemGenes Corporation, catalogue number Macosko201110) were resuspended in lysis buffer to an estimated concentration of 400 beads μL$^{-1}$. Cells and beads were co-encapsulated with oil (QX200™ Droplet Generation Oil for EvaGreen, Biorad) using a microfluidics chip (Part number 3200455, Dolomite). Droplets of around 3 mL of aqueous volume (1.5 mL of cells and beads) were broken with perfluorooctanol in 30 mL of 6× SSC. The harvested beads were then washed twice with 6× SSC and hybridized RNA was reverse transcribed using Maxima H minus Reverse Transcriptase (ThermoFisher). Populations of 2500 reverse-transcribed beads (~100 cells) were separately amplified with 13 cycles of PCR (primers, chemistry, and cycle conditions identical to those previously described) and PCR products were purified with 0.6× AMPure XP beads (Agencourt).

cDNA from an estimated 12,000 E14.5 cells and 8000 P0 cells were pooled, purified and tagmented with Nextera XT DNA Library Preparation kit (Illumina).

**Table 1 Primer sequences**

| Barcoded bead sequence B | 5′ -Bead–Linker-TTTTTTTAAGCAGTGGTATCAACGCAGAGTACJJJJJJJJJJJJJNNNNNNNNTTTTTTTTTTTTTTTTTTTTTTTTTTTTTTT-3′ |
|---|---|
| Template switch oligo | AAGCAGTGGTATCAACGCAGAGTGAATrGrGrG |
| TSO_PCR | AAGCAGTGGTATCAACGCAGAGT |
| P5_TSO_hybrid | AATGATACGGCGACCACCGAGATCTACACGCCTGTCCGCGGAAGCAGTGGTATCAACGCAGAGT*A*C |
| Read 1 custom sequence B | GCCTGTCCGCGGAAGCAGTGGTATCAACGCAGAGTAC |

Input cDNA (1 ng) from each replicate was amplified with custom primer P5_TSO_Hybrid and Nextera index primers (N701, N702, N703, N704, N711, N712, N715, N716, and N718). Tagmented samples were purified twice with 0.6× and 1× AMPure XP beads. All replicates were pooled and sequenced on one Illumina HiSeq 4000 flowcell (eight lanes) to avoid sequencing bias. Read 1 was 20 bp; bases 1–12 represent the cell barcode, bases 13–20 represent the UMI. Read 2 was 50 bp and Read 3 (sample index) was 8 bp. Samples were de-multiplexed using bcl2fastq version 2.18.0.12. Primer sequences used can be found in Table 1.

**Processing of Drop-seq data.** FASTQ files were converted to BAM format, tagged with cell and molecular barcodes, quality-filtered, trimmed, polyA-trimmed, and converted back to FASTQ as previously described[11,12]. Reads were aligned to a mouse-human hybrid genome (mm10-hg19) using STAR[62], then sorted, merged, and exon-tagged as described[11,12]. Bead synthesis errors were corrected as described[12], and BAM files were separated into those containing mouse or human reads. UMIs were determined to be species-specific if >90% of the transcripts came from that species, or considered a doublet if neither species achieved 90% specificity. UMIs were not considered if the transcript count sum (mouse + human) was less than 500. Gene expression matrices were then created using only the mouse-specific UMIs, as described[11,12]. Gene expression matrices were combined from multiple biological replicates; data values for one or more replicates that did not detect a given gene were assigned to zero. Processing steps utilized the Drop-seq Toolkit v1.12 where possible.

**Basic analysis of Drop-seq data.** Cells with fewer than 500 detectable genes or whose mitochondrial contribution exceeded 10% of transcripts were removed, then only genes present in at least 10 cells and having at least 60 transcripts summed across all cells were considered. We then performed batch correction using ComBat[63] where each independent replicate was considered a batch, thus minimizing any technical variation. To reduce the complexity of the data, we performed principal components analysis (PCA) and eigenvalue permutation (500 shufflings) to determine how many principal components (PCs) to use, as described previously[12]. This yielded 89 PCs for E14.5 and 78 PCs for P0. These data were visualized using t-SNE;[18] we iterated both *perplexity* and *learning_rate* parameters to optimize the visualization, ultimately setting these to 50 and 750, respectively. Code made available[12] was used where possible.

**Cluster identification and refinement.** We found that the Louvain-Jaccard clustering method utilized by Shekhar et al.[12] produced highly variable results for our data depending on the number of specified nearest neighbors. We therefore devised an iterative procedure that picks the optimal number of nearest neighbors to use for the given dataset. To do this, we iterate from 10 to 100 nearest neighbors, and for each iteration, repeat the Louvain-Jaccard clustering method. Then, we assess how many clusters formed and how robust they were using silhouette widths[64] based on Spearman correlation distances. After the iteration was complete, we utilized the number of nearest neighbors that produced the maximal average silhouette width across all clusters as a starting point for cell clustering. The silhouette width analysis also allowed us to assess the overall performance of each cluster and demonstrated that the basic clustering method published previously[12] inappropriately assigned many cells to clusters. To refine these cluster assignments, we devised a second iterative approach that attempts to reassign extreme outliers (silhouette width < −0.1) to a better grouping. Over five iterations, these outlier cells of each cluster were given a chance to form their own novel cluster (if its own silhouette width was >0 and there were at least 10 cells) or join the next-best cluster. If a cell was reassigned to the next-best and remained an outlier there, it would be sent back to its original assignment and flagged such that it would not be considered for reassignment in subsequent iterations. This process improved the overall cluster assignments (Fig. 1a) and resulted in the creation of two novel clusters for E14.5, such that the final number of clusters for both E14.5 and P0 was 22. The final cell type assignments were visualized using t-SNE[18] and compared to one another semi-quantitatively using hierarchical clustering with Pearson correlation-based distances.

**Identification of cell type markers and enriched pathways.** Marker genes for each refined cluster were identified as described previously[11,12], and expression summaries were created using the code provided where possible. To identify biological pathways enriched in each cluster, markers whose expression fold-change relative to other clusters exceeding 0 were mapped to human gene symbols and assessed using a hypergeometric test in Piano[65] with MSigDB C2 classifications plus additional neurological gene sets as we described previously[66]. Pathways with an FDR <0.1 and among the top 50 for a given cluster were considered for inclusion in the cell type annotation table.

**Validation by in situ hybridization and immunofluorescence.** Marker genes were validated with in situ hybridization data available on Eurexpress (www.eurexpress.org), Allen Brain Institute (www.developingmouse.brain-map.org) and GENSAT (www.gensat.org). Images were cropped to representative sections of the neocortex, ganglionic eminences, striatum, thalamus and choroid plexus. Marker annotations are provided in Supplementary Data 2.

For immunofluorescence staining, E14.5 and P0 mouse brain were dissected in ice-cold phosphate buffered saline (PBS). After meninges were carefully removed, brains were drop-fixed in 4% PFA overnight at 4 °C. Sagittal brain sections (75 μm) of E14.5 brain were obtained using a Leica VT 1200 vibrotome and stored in PBS at 4 °C. P0 brains were preserved in 30% sucrose and crop-sectioned at 24 μm and stored at −20 °C. E14.5 brain sections were subjected to heat-induced epitope retrieval by steaming in 10 mM citrate buffer for 15 min to achieve robust TBR2 detection. Brain sections were washed briefly in PBS and incubated in blocking buffer (PBS with 2% DMSO, 0.3% Triton-X 100, 2% normal donkey serum and 2% normal goat serum) at room temperature for 1 h. Primary antibodies were diluted in blocking buffer and incubated at room temperature overnight with constant agitation. After three washes with PBS/0.1% Triton X-100, brain sections were incubated with fluorophore-conjugated secondary antibodies at room temperature for 2 h. After three washes with PBS/0.1% Triton X-100, sections were mounted on glass slides using polyvinyl alcohol-based mounting medium supplemented with 0.1% propyl-gallate and DAPI. Primary antibodies used were mouse anti-Reelin (Millipore, MAB5364; IgG1) at 1:1000 dilution, rabbit anti-DKK3 (Abcam, ab2459) at 1:500 dilution, mouse anti-SOX2 (R&D System, MAB2018; IgG2a) at 1:500 dilution, rat anti-TBR2 (eBioscience, 14-4879-12) at 1:500 dilution), rabbit anti-CRE (#13056S, Cell Signaling) at 1:500 dilution and goat anti-CALB2 (Swant, CG1) at 1:500 dilution. Secondary antibodies used were goat anti-mouse IgG1 (Alexa 488 or 647; Thermo Scientific), goat anti-mouse IgG2a (Alexa 488, Thermo Scientific), donkey anti-rabbit IgG (Alexa 488, 568 or 647, Thermo Scientific), donkey anti-goat IgG (Alexa 568, Thermo Scientific) and donkey anti-Rat IgG (Cy3, Jackson ImmunoResearch). All secondary antibodies were diluted 1:1000 in blocking buffer. Images were acquired using a Zeiss LSM 780 laser scanning confocal microscope with a Plan-Apochromat 40 ×/1.4 Oil DIC objective.

**Cell type sub-clustering method.** To identify genes whose expression pattern was heterogeneous within a cluster, we required that a given gene was detected in at least 25% of cells but not more than 75% of cells within a cluster, then further refined the heterogeneous gene list using a feature selection tool[67] on the expression of all cells within the cluster. The expression of these genes in all cells for that cluster were then clustered hierarchically and inspected manually for coherent patterns representing sub-clusters. Genes within each sub-cluster were then assessed for functional enrichments using ToppGene[68].

**Pseudotiming analysis.** For cells from Cluster 21-E or the $Reln^+$ or Precursor-like sub-clusters of Cluster 17-E, we performed dimensionality reduction using PCA on Z-scored gene expression data, for only those genes with abundance of at least 2 TPM in at least 10 cells. We then created a curvilinear trajectory and ordering of these cells using Slingshot[69], with Cluster 21-E and $Reln^+$ sub-clusters constrained as endpoints.

**Comparisons to other published datasets.** Gene expression data for Tasic et al. adult mouse cortex[22] was obtained from GEO (GSE71585) and collapsed within cell types by taking the mean of all cells. Data were similarly obtained for Zeisel

et al. adult mouse cortex[35] from GSE60361 and collapsed using Level 2 cell type labels. Data were similarly obtained and handled for Molyneaux et al. from GSE63482[7]. A common set of marker genes was defined based on the intersection of published gene lists[22,35] with our cell-type- (log-fold-change >1.5) and subcluster-specific genes. Expression data for each dataset were then subsetted based on this common gene list, and expression patterns were compared pairwise for all cell types using Spearman correlation. These correlations were then illustrated as a circos plot, where lines were drawn only for comparisons between data from this study and other studies, and when those correlations exceeded a minimum threshold. Cross-species analyses were performed in a similar fashion using recently published data from human fetal (7–11.5 pcw) tissue[39]. All human marker genes were converted to mouse orthologs using HGNC approved conversions.

**Cross-age comparisons and disease gene subtyping**. Expression data for each cell cluster was merged by taking the mean of all cells, and all clusters from both ages were compared to each other using Pearson correlations. To focus on genes most responsible for cell-type-specificity, we used genes that were identified either as cell type markers (log-fold-change >1.5) or among those that were included in the sub-cluster analysis. Lists of commonly mutated genes for each disease of the cortex were downloaded as follows: ALS (ref. [47], Tables 1, 2), Alzheimer's disease (ref. [48], Table 1 + APP, PSEN1, PSEN2), ASD (SFARI Gene classifications Syndromic and Class 1, obtained August 22, 2017), ciliopathies[51,52], and schizophrenia[50]. For each disease gene set, we performed hierarchical clustering of median-centered expression values across all 44 combined clusters (E14.5 and P0 included) using 1-Pearson correlation distance. We then cut the gene dendrograms at a height of 1.5 to divide genes into subgroups (except ASD, where we instead specified a height of 1.25). Then for each gene subgroup (#genes ≥3), we collapsed expression across related cell types by taking the median across those cell types within an age (e.g., median of all E14.5 interneurons) to obtain 15 total values per gene subgroup (layer-specific, interneuron, progenitor-like, and each non-neuronal cell type for each age). The statistical association between gene expression subtypes and clinical phenotypes was determined using a two-sided Fisher's Exact Test, using Monte Carlo simulation of $p$-values (10,000 replicates).

**Code availability**. All code for data analysis, including cluster identification and refinement, cellular sub-clustering analysis, as well as processed and unprocessed gene expression matrices are available on GitHub at https://github.com/jeremymsimon/MouseCortex.

**Reporting summary**. Further information on experimental design is available in the Nature Research Reporting Summary linked to this article.

## Data availability

The web-based visualization tool described above is available at http://zylkalab.org/data. Raw and processed data were also deposited to the Gene Expression Omnibus under accession 'GSE123335'.

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

## Acknowledgements

We thank the High Throughput Sequencing Facility at the University of North Carolina for library sequencing, and Hyejung Won and Jason Stein for discussion of disease subtyping analysis, Marcus Basiri and Garret Stuber for helping to establish the Drop-seq microfluidics station and discussion of Drop-seq analyses, and Hongwei Liu for creation of web-based data visualization tools. This research was supported by the National Institute of Environmental Health Sciences of the National Institutes of Health (DP1ES024088, R56ES028236, M.J.Z.) and by the Simons Foundation (Award ID # 393316, M.J.Z.). E.S.A. and J.G. were supported by the National Institute of Neurological Disorders and Stroke (NINDS, NS090029). J.M.S was supported by The Eunice Kennedy Shriver National Institute of Child Health and Human Development (U54HD079124) and NINDS (P30NS045892). J.K.N. was supported by NINDS (F31NS105397).

## Author contributions

M.J.Z, L.L. and J.M.S. designed the study. E.S.M. performed cortical dissections. L.L. performed Drop-seq, prepared libraries for high-throughput sequencing and annotation of marker genes. J.M.S. analyzed all Drop-seq data. L.X. performed immunohistochemistry and provided technical assistance. J.K.N. provided technical assistance. J.G. and E.S.A. provided assistance with ciliopathy phenotype linkages. L.L., J.M.S., and M.J.Z. wrote the manuscript.

## Additional information

**Competing interests:** The authors declare no competing interests.

