## [Peer Review File · Nature Communications]

Reviewers' comments:

Reviewer #1 (Remarks to the Author):

The authors do a seeming competent analysis of cortical development outlining developmental clustering in developing cortical cells using single cell RNA-seq methods. Using a comparatively small number of cells they analyze 10K cells at E14.5 and a addition 7.5K at P0. The authors identify both a plethora of genes suggesting that their methods are robust but the true rigor of the approach is impossible to assess without some further basic information. Specifically, no where in the manuscript could I find either the gene cutoff threshold for genes per cell or the median UMI counts per cell. Without this information, it is hard to take seriously what the level of depth that was achieved. In addition, the methods for comparing gene expression across groups seemed anecdotal. Technical error can make such comparisons difficult and I would refer the authors to recent methods available on Bioarxiv for creating a manifold to comparison of common variance. In addition deconvolving developmental differences attributable to state of maturation compared to cell type are a typical source of confounding error and the authors failure or inability to link cell types observed to those observed in the adult clearly compromises the power of their analysis. With more information and detail, it remains possible that the work would prove suitable for publication but in its present form the paper is underdeveloped and needs considerably further work.

Reviewer #2 (Remarks to the Author):

Loo et al. present their study of mouse cortical development based on single cell RNA-seq. They profile 2 time points E14.5 and P0, identify major cell types and/or states and connect some of the defined types to neurological diseases through previously identified marker genes.

Although I have little doubt that these data are meaningful and valuable, the presentation and interpretation are very basic and many conclusions are not presented in a scholarly and thoughtful fashion.

Several points to illustrate the issues:

1. What is the reason that not all clusters were presented at the same time – but in the two step-fashion (with second step – Fig 2 – being very unusual way to present subtypes). How is a reader supposed to get a full picture of the dataset? It is a common to do clustering iteratively, but then, it is best to present all final clusters together. Fig 2 focuses on proportions – why not trees or tSNEs? It is an unusual and not efficient way to represent subtypes/subclusters.
2. Which clusters are known, and which are new? Which single cell transcriptomic papers have already identified some of these types – there are developmental studies of non-neuronal cells, for example. Can that be summarized?
3. How do these types relate to adult transcriptomic types previously described in the cortex?
4. “Thus, cellular expression profiling has the potential to identify novel disease subtypes and cellular vulnerabilities associated with brain diseases.” Do we have any reason to think that these different genes in fact yield different disease subtypes? Is there evidence for that – if yes please cite that. Or are they just different contributors to the disease phenotype, but may not actually result in different disease types? I think it would also be scholarly to cite previous work that has used single cell transcriptomics to identify cell types that may be relevant for certain diseases, like Campbell et al, Nat Neu, 2017 (arcuate hypothalamus scRNA-seq study).
5. “In conclusion, we found that the unfolding cellular complexity of brain development— formerly pieced together over decades of research—can be reconstructed with a single experimental approach.” – This is simply not true. Single cell transcriptomics is a great starting point, but for example, precursor-progeny relationships, which are essential for understanding development are very difficult to establish without additional experimental approaches, for example, lineage tracing.

6. Fig 1a: How was the dendrogram generated? What is normalized expression – normalized to what? Is mean or median gene expression per cluster represented?
7. Fig 1b legend: “Fractional proportions of cortical cells, averaged across all biological replicates (n=6, E14.5; n=3, P0). Non-cortical cells were excluded.” How does this compare to what we know about the proportions of cellular classes? Any answer is fine, but it should be discussed at least. It is known that not all cells survive these procedures equally well.
8. Fig 1c legend: how were these distributions determined?
9. Fig 3 should include at least one of the examples from the related supplemental figs to show how the information in this summary table was derived. The legend is not helpful.
10. Could the same conclusions (in Fig 3) be derived from already existing adult single cell RNA-seq cortical data?

Reviewer #3 (Remarks to the Author):

The manuscript by Loo et al., provides an important dataset of single cell gene expression during mouse cortical development. A catalog of cell-type-specific gene expression during brain development could serve as a resource for exploring the vulnerability of distinct cell populations as well as for understanding molecular factors regulating fate specification during brain development.

Technical aspects of the experimental design are well thought out. For example, the authors use 9 biological replicate samples and inhibit ion channels to block immediate early gene activation during cell dissociation. Similarly, the authors innovate on single cell clustering methods by updating the Louvain-Jaccard clustering method to identify an optimal K value for clustering, which was arbitrary in the original Shekhar paper. In addition, the authors thought deeply about the interpretation of distinct transcriptomic clusters to provide a curated interpretation of single cell clusters, along with many interesting heatmaps of genes carefully curated from the literature.

However, I have some major concerns. Overall, the paper felt like a high-quality figure 1 that was still missing deeper analysis or novelty beyond resource generation. Specifically:

1) Why are only two timepoints selected? The paper title implies the paper is a catalog of mouse cortical development, and E14.5 and P0 are a fine choice if limited to two timepoints, but if the paper is to serve as a resource or catalog, two timepoints may not be sufficient. Otherwise, the paper should further prove some element of novelty like unappreciated heterogeneity.

2) All validation comes from images from the Allen Brain Institute or other databases. These results certainly support the interpretations and the quality of the dataset, but claims of distinct clusters and unexpected subtypes require validation of co-expression patterns for cluster marker genes and/or reciprocal expression of markers for distinct cell types. For example, L5 and L6 are quite heterogeneous in mouse. The authors find multiple sub-clusters assigned to distinct cortical layers by marker gene/in situ database interpretation, but what makes these sub-clusters different? Is the difference simply neuronal maturation, or is there early evidence for fate refinement into heterogeneous cortical subtypes? This level of cluster validation would require FISH or in situ + immune to show the co-expression patterns occur in distinct cell populations. Similarly, the authors find an apparent differentiation trajectory among Cajal-Retzius neurons suggesting *Rspo3*, *Fabp7* and other genes may transiently precede *Reln* expression, but the authors do not show this for example by co-staining for both *Rspo3* and for *Reln* with a pan Cajal-Retzius lineage marker.

The authors do not need to validate every sub-cluster, but certainly demonstrating a few claims of unappreciated heterogeneity would help to support the level of granularity achieved by iterative

clustering and further in Figure 2.

3) One major rationale for studying cortical development is to relate developmental cell types to adult cell types. How do the clusters observed here relate to those from recent studies of adult mouse brain (e.g., Tasic, Linnarrson). For example, Tasic finds 19 distinct glutamatergic cell types, and a new biorxiv studies suggests even more diversity across areas (<https://www.biorxiv.org/content/early/2017/12/06/229542>). Is there evidence for this diversity already at P0?

4) How do the data compare with Molyneaux 2015 (DeCon) which purified cell types of mouse developing cortex for bulk transcriptome

5) The additional heterogeneity in Figure 2 is very nice to see. For example, breaking 21p into meningeal and pericyte subtypes is very useful. I am surprised that the iterative clustering did not pull out these cell type distinctions. If the paper is to be framed as a resource, it would be useful to have marker genes for each broad type as well as for each subtype.

Minor points

-Figure 1A should indicate the number of cells in each cluster, perhaps next to the length correlation column

-At P0 nearly all cortical neurons have been generated, but lamination is not complete, so it is overstating in the text that all layers have formed

-Cells with oRG behavior have also been reported in mice, but are more rare and do not occupy a distinct histological compartment

We thank the reviewers for their careful critiques. We addressed each of these concerns below and believe the suggestions made here significantly improved our manuscript. Further, we made additions and modifications to the figures and format of the manuscript, as well as created a web-based interface to visualize our data (<http://zylkalab.org/data>; reviewer password is “lipin”). We also released our detailed, fully documented code and raw data on GitHub (<https://github.com/jeremysimon/MouseCortex>) to aid those wishing to reproduce our results or apply our new methods to novel datasets.

The reviewers noted below that our single-cell catalog could be an extremely “meaningful and valuable”^{R2} resource for “exploring the vulnerability of distinct cell populations as well as understanding molecular factors regulating fate specification during brain development.”^{R3} We believe these tools and code will significantly broaden the utility of our dataset and methodologies, and serve as a common reference for ongoing and future neurodevelopmental research.

Reviewer #1 (Remarks to the Author):

The authors do a seeming competent analysis of cortical development outlining developmental clustering in developing cortical cells using single cell RNA-seq methods. Using a comparatively small number of cells they analyze 10K cells at E14.5 and a addition 7.5K at P0. The authors identify both a plethora of genes suggesting that their methods are robust but the true rigor of the approach is impossible to assess without some further basic information. Specifically, no where in the manuscript could I find either the gene cutoff threshold for genes per cell or the median UMI counts per cell. Without this information, it is hard to take seriously what the level of depth that was achieved.

We now include the median number of reads, genes, and transcripts per cell as part of our modified Figure 1. As we note in our modified Results section, our depth of ~12,000 reads per cell is on par with or exceeds other similar studies. Additionally, we have edited our Methods section to include the threshold we employed for minimum number of detected genes per cell; this was set to 500, consistent with similar recent studies.

In addition, the methods for comparing gene expression across groups seemed anecdotal. Technical error can make such comparisons difficult and I would refer the authors to recent methods available on Bioarxiv for creating a manifold to comparison of common variance.

To address, we limited our analyses to either A) quantitative comparisons across identified cell types within a timepoint, as many recent similar studies have employed (ie constructing a cell type “atlas” based on clear cell-type-specific expression patterns); or B) semi-quantitative comparisons across timepoints, where we exclusively rely on correlation-based metrics not affected by these confounds. Since we are not making any claims with respect to lineage tracing or perform any other direct quantitative comparative analysis of our data across timepoints, we feel our methodology is sound.

In addition deconvolving developmental differences attributable to state of maturation compared to cell type are a typical source of confounding error and the authors failure or inability to link cell types observed to those observed in the adult clearly compromises the power of their

analysis. With more information and detail, it remains possible that the work would prove suitable for publication but in its present form the paper is underdeveloped and needs considerably further work.

The suggestion to link our cell types to those observed in the adult was shared among all three reviewers, and we agree that it strengthens our claims on the identity of each of these identified cell types as well as the utility of the dataset as a whole. We therefore pursued this comprehensively and rigorously, and added Supplemental Figure 14 as well as a dedicated section in both the Results and Discussion. Briefly, we performed correlation-based comparative analyses between the gene expression patterns of our cell types and that of Zeisel *et al* (single-cell, Fluidigm, adult mouse cortex and hippocampus), Tasic *et al* (FACS-sorted cell populations + sequencing, adult mouse cortex), and Molyneaux *et al* (“DeCoN”; FACS-sorted cell populations + sequencing, developing mouse cortex). We chose these three studies because they were similar in methodology (e.g. Zeisel), similar in age (e.g. Molyneaux), or different in methodology and age (e.g. Tasic). Overall, our cortical excitatory neurons (Layers II-VI) correlate most closely with that of the Molyneaux DeCoN data, as expected due to the similar developmental age of their neurons. We also observed correlation between our Int2 class of interneurons and Sst⁺ interneurons of the adult cortex.

To strengthen these similarities further, we performed a similar analysis of the recent Nowakowski *et al* study of the developing human brain. We focused on 7-9pcw and 20-23pcw as approximate equivalents in age to our E14.5 and P0 cells, based on Workman *et al* (PMID 23616543). Due to the nature of the samples they obtained, their number of cells per cell type was small. We observed correlation between several radial glia/precursor populations, excitatory neurons, and several glial cell types at both ages, as described in our revised manuscript.

Together, these data demonstrate the transcriptional similarities between our identified cell types and those from other related datasets and strengthen the utility and power of our study.

Reviewer #2 (Remarks to the Author):

Loo *et al.* present their study of mouse cortical development based on single cell RNA-seq. They profile 2 time points E14.5 and P0, identify major cell types and/or states and connect some of the defined types to neurological diseases through previously identified marker genes.

Although I have little doubt that these data are meaningful and valuable, the presentation and interpretation are very basic and many conclusions are not presented in a scholarly and thoughtful fashion.

We originally wrote this manuscript as a brief communication to *Nat. Neurosci.* and were advised to transfer it directly to *Nat. Comm.* We agree that this pared down brief communication format made it difficult to fully describe what we did, what we found, and to present our work in a scholarly fashion. We have now reformatted as a traditional manuscript, with an academically rigorous introduction and discussion.

Several points to illustrate the issues:

1. What is the reason that not all clusters were presented at the same time – but in the two step-fashion (with second step – Fig 2 – being very unusual way to present subtypes). How is a reader supposed to get a full picture of the dataset? It is a common to do clustering iteratively, but then, it is best to present all final clusters together. Fig 2 focuses on proportions – why not trees or tSNEs? It is an unusual and not efficient way to represent subtypes/subclusters.

We have modified and essentially combined Figures 1 and 2 to address this concern. Our modified Figure 1 shows the data more globally, including tSNEs of the cell types for each timepoint, as well as expression of major marker genes in tSNE coordinates. The modified Figure 2 shows more details about each cluster, the relatedness across timepoints (via correlation-based metrics to draw the dendrogram), the number of cells, the number of sub-clusters identified, as well as sub-cluster proportions and specific expression patterns for a selection of populations.

Other groups are now recognizing the importance of subclustering single cell data (see for example, PMID: 29466745). Initial clusters must be subclustered to fully identify unique cell subtypes, as we have done.

2. Which clusters are known, and which are new? Which single cell transcriptomic papers have already identified some of these types – there are developmental studies of non-neuronal cells, for example. Can that be summarized?

To our knowledge, our dataset is the first to comprehensively identify each neuronal and non-neuronal cell type in the developing brain. The ISH images detailed in the Supplementary Materials (and exhaustively summarized in Supplementary Table 3), as well as the newly added comparative analyses described above with other adult and developing mouse and human datasets show the relationship between the cell types we identified and those identified in previous single cell studies (at adult ages). Further, we describe and validate a novel class of Cajal-Retzius precursor cells; more details can be found below.

3. How do these types relate to adult transcriptomic types previously described in the cortex?

The suggestion to link our cell types to those observed in the adult was shared among all three reviewers, and we agree that it strengthens our claims on the identity of each of these identified cell types as well as the utility of the dataset as a whole. Please see our response to Reviewer 1 above, as well as Supplemental Figure 14, for our characterization of comparisons between this and other datasets.

4. “Thus, cellular expression profiling has the potential to identify novel disease subtypes and cellular vulnerabilities associated with brain diseases.” Do we have any reason to think that these different genes in fact yield different disease subtypes? Is there evidence for that – if yes please cite that. Or are they just different contributors to the disease phenotype, but may not actually result in different disease types? I think it would also be scholarly to cite previous work that has used single cell transcriptomics to identify cell types that may be relevant for certain diseases, like Campbell et al, Nat Neu, 2017 (arcuate hypothalamus scRNA-seq study).

As the reviewer points out, there are previously published studies identifying cell types that are affected in Alzheimer's disease, or other conditions such as food deprivation (Campbell et al, among others). This is now mentioned and cited in our revised manuscript. However, for ASD, ALS, Alzheimer's disease, and schizophrenia, there is insufficient evidence to evaluate the extent to which different gene mutations produce specific disease phenotypes. We therefore sought to find another heterogeneous class of neurodevelopmental disorders that has documented genotype-phenotype associations. Ciliopathies, such as Bardet-Biedl Syndrome, Joubert Syndrome, etc, are pleiotropic and genetic abnormalities can manifest in a variety of tissues including the brain. Links between gene mutations and neurodevelopmental defects have been published (PMID 28698599, 26206566). We applied our same methodology to group these ciliopathy-linked genes into subtypes based on their cell-type-specific gene expression patterns in the developing cortex, and identified four subtypes. These subtypes were characterized by expression primarily in the choroid plexus (subtype 1), proliferative cells (subtype 2), neurons (subtype 3), and glia (subtype 4). We hypothesized that these subtypes might therefore exhibit differences in prevalence of microcephaly, hydrocephaly, axonal tract defects, or intellectual disability. We tabulated the number of genes in each expression subtype that were associated with each clinical phenotype, and indeed found a statistically significant association suggesting that these clinical phenotypes were distributed non-randomly among the four subtypes identified (Fisher's exact test, $p=0.0406$). These data suggest that expression-based genetic subtyping could have prognostic power.

We have included these new results in the modified Figures 4-5, and discussed this further in our Discussion section.

5. "In conclusion, we found that the unfolding cellular complexity of brain development—formerly pieced together over decades of research—can be reconstructed with a single experimental approach." – This is simply not true. Single cell transcriptomics is a great starting point, but for example, precursor-progeny relationships, which are essential for understanding development are very difficult to establish without additional experimental approaches, for example, lineage tracing.

We agree that additional experiments would be necessary to validate any putative lineage from single-cell transcriptomics, and have removed this sentence but included mention of emerging methodologies better suited to address these questions in the Discussion section.

6. Fig 1a: How was the dendrogram generated? What is normalized expression – normalized to what? Is mean or median gene expression per cluster represented?

The dendrogram was created by first taking the gene-wide mean of all expression values within a cluster, where those expression values were already scaled, centered, and batch-corrected, as described. Then, we focused only on the genes we determined to be cluster-specific or sub-cluster-specific markers. We then compared the expression patterns of those genes across each cell type and timepoint using Pearson correlation as a distance metric. This is now included in our Methods section.

7. Fig 1b legend: “Fractional proportions of cortical cells, averaged across all biological replicates (n=6, E14.5; n=3, P0). Non-cortical cells were excluded.” How does this compare to what we know about the proportions of cellular classes? Any answer is fine, but it should be discussed at least. It is known that not all cells survive these procedures equally well.

The estimated proportions of cellular classes is described as follows in our modified Results section (note this panel has now moved into the Figure 2 key):

“Nearly 40% of cells at E14.5 were progenitor-like, including multiple classes of radial glia (RG) and intermediate progenitors localized to the ventricular zone (VZ) and sub-ventricular zone (SVZ), but represented only 28% of cells by P0, as expected². The ratio of excitatory to inhibitory neurons was in line with previous estimates that range from 2:1 to 5:1 (Refs. ^{3,4}). Additionally, the P0 cerebral cortex contained a greater proportion of non-neuronal cells relative to the E14.5 cortex, which is consistent with the known timing of glial proliferation²”

8. Fig 1c legend: how were these distributions determined?

This figure panel has been removed from our modified manuscript

9. Fig 3 should include at least one of the examples from the related supplemental figs to show how the information in this summary table was derived. The legend is not helpful.

We updated the figure legend for this figure to be more helpful, and inserted the full ciliopathy heatmap (formatted just like the other disease Supplemental Figures) as Figure 4, as requested, to better describe how the distilled summary was derived.

10. Could the same conclusions (in Fig 3) be derived from already existing adult single cell RNA-seq cortical data?

Given that the existing adult single-cell datasets by their nature lack progenitor cell populations—one of the four primary meta-cell-types we focus on in this figure—and that gene expression in the adult cortex is similar but not identical to that of the developing cortex, it is difficult to evaluate whether this type of analysis yields similar conclusions. Nonetheless, we employed the same methodology on the Tasic adult cortex dataset, focusing on genes mutated in ASD and ALS. We observed the same number of gene clusters (6 in ASD, and 2 in ALS), presented below. Since the gene composition was similar but not identical to that of our developing cortex cell types, we have modified our Discussion section to make clear that cell-specific expression patterns but also age (and likely sex) are important factors in determining the effects of these mutations on the brain.

ASD genes

Reviewer #3 (Remarks to the Author):

The manuscript by Loo et al., provides an important dataset of single cell gene expression during mouse cortical development. A catalog of cell-type-specific gene expression during brain development could serve as a resource for exploring the vulnerability of distinct cell populations as well as for understanding molecular factors regulating fate specification during brain development.

Technical aspects of the experimental design are well thought out. For example, the authors use

9 biological replicate samples and inhibit ion channels to block immediate early gene activation during cell dissociation. Similarly, the authors innovate on single cell clustering methods by updating the Louvain-Jaccard clustering method to identify an optimal K value for clustering, which was arbitrary in the original Shekhar paper. In addition, the authors thought deeply about the interpretation of distinct transcriptomic clusters to provide a curated interpretation of single cell clusters, along with many interesting heatmaps of genes carefully curated from the literature.

Thank you for astutely identifying novel aspects of our work.

However, I have some major concerns. Overall, the paper felt like a high-quality figure 1 that was still missing deeper analysis or novelty beyond resource generation.

We originally wrote this manuscript as a brief communication to Nat. Neurosci. and were advised to transfer it directly to Nat. Comm. This pared down brief communication format made it difficult to fully describe what we did, what we found, and to present our work in a scholarly fashion. We have now reformatted as a traditional manuscript, added a high quality Fig. 1 as you recommended, and added an academically rigorous introduction and discussion. We also developed a web-based interface for data visualization and released our code, to demonstrate the utility of these data and analytical approach to the community.

Specifically:

1) Why are only two timepoints selected? The paper title implies the paper is a catalog of mouse cortical development, and E14.5 and P0 are a fine choice if limited to two timepoints, but if the paper is to serve as a resource or catalog, two timepoints may not be sufficient. Otherwise, the paper should further prove some element of novelty like unappreciated heterogeneity.

Single cell sequencing is a new technology and is costly, requiring that we focus our studies. Since this technology has not previously been applied to the developing mouse cortex, we focused on two key times of corticogenesis—E14.5, representing a stage in which upper-layer neurons are just beginning to form and progenitor cell populations are still proliferating, and P0, when neurons corresponding to all six cortical layers have been born and gliogenesis has begun. We now state this in the manuscript. To further provide a resource to the community, we now provide a user-friendly web interface so that gene expression in each cell type can be determined (see zylkalab.org/data, and password information above).

2) All validation comes from images from the Allen Brain Institute or other databases. These results certainly support the interpretations and the quality of the dataset, but claims of distinct clusters and unexpected subtypes require validation of co-expression patterns for cluster marker genes and/or reciprocal expression of markers for distinct cell types. For example, L5 and L6 are quite heterogeneous in mouse. The authors find multiple sub-clusters assigned to distinct cortical layers by marker gene/in situ database interpretation, but what makes these sub-clusters different? Is the difference simply neuronal maturation, or is there early evidence for fate

refinement into heterogeneous cortical subtypes? This level of cluster validation would require FISH or in situ + immune to show the co-expression patterns occur in distinct cell populations.

The varieties of Layer V–VI neurons could be discriminated by a combination of factors, either indicative of their underlying function and/or characteristic of regional specificity. We have modified this section of our Results as follows to address this more comprehensively:

“All E14.5 excitatory neuron clusters (5-E, 13-E, 3-E, 7-E and 2-E) broadly expressed *Bcl11b*, a deep layer marker²² (**Supplementary Fig. 4 and 6**). Layer V–VI neurons could be further distinguished based on expression of genes characteristic of their function or that demonstrate regional specificity. For example, Clusters 5-E and 13-E both expressed *Fezf2*, which is normally expressed at high levels in Layer V SCPN and lower levels in Layer VI CThPN²³. These two clusters could be further segregated spatially by expression of *Crym*, which is expressed more caudally²⁴, and *Mc4r*, which is expressed more rostrally²⁴ (**Supplementary Fig. 4 and 6 and Supplementary Table 3**). Cluster 7-E also showed regional specificity, given its expression of *Tfap2d*, which is expressed more rostrally²⁵ (**Supplementary Fig. 4 and 6**). We also identified three classes of Layer II–IV (upper-layer) neurons in the P0, but not E14.5, cortex, consistent with the later birthdate of upper layer neurons. Each of these clusters (Clusters 1-P, 4-P, 15-P) expressed *Satb2* and *Pou3f1*, and 4-P were further specified by expression of *Nrgn*, *Inhba* and *Pvrl3* (**Supplementary Figure 10**).”

We did not detect similar regional heterogeneity among interneuron classes. This is consistent with data from the adult cortex demonstrating regional specificity in excitatory neuron clusters (<https://www.biorxiv.org/content/early/2017/12/06/229542>). There is also some evidence for this regional specificity to be a function of neuronal maturation/birth timing; for example, *Tfap2d* is expressed more rostrally at E14.5 but much more broadly by E18.5. Taken together, our data do not necessarily suggest early fate refinement, but rather that transcriptionally similar cell populations co-occupy the same laminar space but reside in regionally distinct locations.

Similarly, the authors find an apparent differentiation trajectory among Cajal-Retzius neurons suggesting *Rspo3*, *Fabp7* and other genes may transiently precede *Reln* expression, but the authors do not show this for example by co-staining for both *Rspo3* and for *Reln* with a pan Cajal-Retzius lineage marker.

The authors do not need to validate every sub-cluster, but certainly demonstrating a few claims of unappreciated heterogeneity would help to support the level of granularity achieved by iterative clustering and further in Figure 2.

Regarding the apparent differentiation trajectory among Cajal-Retzius cells, we performed immunofluorescence as the reviewer suggested, and also added results describing pseudotiming of this trajectory using gene expression data. In regard to the staining and validation of this trajectory, *Rspo3* stained cells in a punctate manner (data not shown), so we instead opted to stain for *DKK3*, *EOMES/TBR2*, *SOX2*, and *RELN*. As we describe in the modified Results section, *DKK3* and *RELN* protein were detected in both cortical hem and Layer I cells. Intriguingly, a gradient of these marker proteins

was apparent in the cortical hem, delineating a differentiation trajectory from cortical hem progenitors, many of which expressed *Dkk3* but not *Reln*, to mature Cajal-Retzius cells, the majority of which expressed *Reln* but not *Dkk3*. Given that Cajal-Retzius cells can originate from the cortical hem, our data, together with the new pseudotiming analysis, suggest that this sub-cluster of cells represents precursors destined to migrate to Layer I. Our new Figure 3 presents these IF data as well as the transcript-level pseudotiming results to illustrate this differentiation trajectory.

3) One major rationale for studying cortical development is to relate developmental cell types to adult cell types. How do the clusters observed here relate to those from recent studies of adult mouse brain (e.g., Tasic, Linnarrson). For example, Tasic finds 19 distinct glutamatergic cell types, and a new biorxiv studies suggests even more diversity across areas (<https://www.biorxiv.org/content/early/2017/12/06/229542>). Is there evidence for this diversity already at P0?

The suggestion to link our cell types to those observed in the adult was shared among all three reviewers, and we agree that it strengthens our claims on the identity of each of these identified cell types as well as the utility of the dataset as a whole. Please see our response to Reviewer 1 above, as well as Supplemental Figure 14, for our characterization of comparisons between this and other datasets.

4) How do the data compare with Molyneaux 2015 (DeCon) which purified cell types of mouse developing cortex for bulk transcriptome

Please see our response to Reviewer 1 above, as well as Supplemental Figure 14, for our characterization of comparisons between this and other datasets.

5) The additional heterogeneity in Figure 2 is very nice to see. For example, breaking 21p into meningeal and pericyte subtypes is very useful. I am surprised that the iterative clustering did not pull out these cell type distinctions. If the paper is to be framed as a resource, it would be useful to have marker genes for each broad type as well as for each subtype.

The result of our sub-clustering analysis has now been combined with the original Figure 1 panel to simplify the presentation of clusters and sub-clusters. Where one draws the line between being a “cell type”, “cell state”, “cell sub-type”, etc is a very interesting question and emerging debate, that we now discuss (in the Discussion section).

While we can't rule out the possibility that sequencing more cells would cause our clustering methodology to separate these into distinct parts, as we mention in the revised text, others are finding it necessary to subcluster to resolve additional cell types or states. While these studies were published subsequent to our initial submission, they now support our iterative approach for analyzing single cell seq data.

We revised Figure 2 and associated Supplementary Figures 16-17 to show annotated marker genes for discriminating these sub-clusters of cells.

Minor points

-Figure 1A should indicate the number of cells in each cluster, perhaps next to the length correlation column

We added this to our modified version of Figure 2

-At P0 nearly all cortical neurons have been generated, but lamination is not complete, so it is overstating in the text that all layers have formed

We have modified this sentence to read "...when neurons corresponding to all six cortical layers have been born"

-Cells with oRG behavior have also been reported in mice, but are more rare and do not occupy a distinct histological compartment

We have modified this sentence to read "Primates have an additional form of RG known as outer RG..."

Reviewers' comments:

Reviewer #3 (Remarks to the Author):

The manuscript by Loo et al., has improved upon revision. The Drop-seq data are high quality, the clustering/sub-clustering methods for both ages are sound, and the cluster interpretations are well reasoned (with a few possible exceptions below). Most of the tissue validation is still based on the expression of single genes, rather than proving co-expression patterns, but the authors do provide staining to validate co-expression patterns during the differentiation and migration of Cajal-Retzius neurons from the cortical hem. Like many recent single cell papers, the text necessarily includes a tour of cluster and defense of interpretations/disease implications, but I think there is potential to also connect more to recent themes cortical development.

I still have several comments that the authors can likely address:

- 1) Are the authors suggesting that 5-E and 13-E represent area-specific subsets of Fezf2 neurons? If so, this should be made more explicit and analyzed more deeply. Nowakowski et al., 2017 recently showed that in human, area-specific excitatory neurons emerge early in cortical development prior to sensory experience. The author should reference this work and address whether the same markers delineate these subclusters. In addition, both the Tasic paper in adult mouse, and the Nowakowski paper in developing human cortex sampled extreme poles of the cortex, leaving open the question of whether there are discrete types or whether the excitatory neurons vary along a gradient. In this study, was the whole rostrocaudal extent of the brain sampled? And could 5-E and 13-E neurons be better represented along a gradient? Also, why is there only a single Fezf2 cluster at the P0 timepoint – in the Tasic paper, it looks like areal differences may be greater in adult? Does this cluster contain neurons that appear more rostral or caudal? Finally, do the upper layer excitatory neurons also appear to be area-specific. To summarize, if the authors are advancing the possibility of area-specific excitatory neurons, which has recently emerged in the field, there needs to be more analysis and comparison with other findings.
- 2) The 3-E cluster is called Layer 5, but expresses high levels of Satb2, Pou3f1, and Cux2, no Fezf2, and lower levels of Bcl11b and Tbr1. These might be immature upper layer neurons or a mature intratelencephalic cluster. As the authors note later in the text, upper layer neurogenesis does start around E14.5. In addition, neuronal maturation marker are lower in this cluster, so my guess is that these are indeed immature upper layer neurons.
- 3) Cluster 8P is surprising in that it contains markers of excitatory neurogenesis (Eomes, Neurod6) along with Calb2. I believe that Calb2 is not usually observed in excitatory neurons, except for Cajal-Retzius cells, but this is interpreted as an SVZ cluster and captured at a stage when we might not expect new Cajal-Retzius neurons to be generated. The in situ data from Allen provided in the supplement shows Calb2 most strongly staining marginal zone. If the authors could provide co-expression data from SVZ of Neurod6 (or other excitatory lineage markers) and Calb2, this would be helpful in confirming this unexpected population. Alternatively, 8P could represent doublets between Cajal-Retzius and SVZ progenitors (2-P), or there is another interpretation I am unaware of.
- 4) Why are 10P and 13P considered astrocyte IPCs when they do not express cell cycle markers? Would it not make more sense to consider these astrocytes? Also, the Aqp4 in situ in S12 seems to not show expression.
- 5) I am surprised that the SST/Lhx6 interneurons don't translate well across species or age. Nowakowski et al., reported both CGE and MGE-derived inhibitory neurons in cortex, and markers like SST, Lhx6, and Nkx2-2 are likely conserved. Have other genes changed to an extent to limit homology, or is this a limitation of the correlation method? (The visualization of correlations between different datasets is very nice)
- 6) The timepoints comparing E14.5 mouse to human are not well aligned and must be changed. E14.5 mouse corresponds to PCW12.5 to PCW14.5, when deep layer neurogenesis starts to wind down in human. PCW7-9 is much too early. This tool provides one method for conversion:

<http://translatingtime.org/translate>

7) vRG probably is meant to refer to ventricular radial glia, rather than radial glia in ventral telencephalon

8) In figure 5, does layer-specific mean excitatory neuron?

9) I would probably combine figure 4 and 5 as Figure 4 is an example that is generalized (and included) in Figure 5

10) The web tool is great – the dataset and interpretations will be more valuable for distributing the data in this way. Can you also let users pick a cluster and get all the cluster markers? Also, can you allow a user to input a list of genes and get table of values? Alternatively, the supplement is fine for this purpose.

11) I appreciate the supplementary table linking each interpretation to genes and literature searches or in situs.

Reviewer #4 (Remarks to the Author):

This manuscript by Loo et al. makes use of single cell RNA-seq to generate a transcriptomic description of cell types in the developing mouse cerebral cortex at E14.5 and P0. Validation of identified clusters is done through ISH and IHC.

Overall, the authors have substantially revised the manuscript in response to the prior review comments. Significant improvements include: the development of a web-based interface for data visualization, improved consideration of other studies using similar methodologies and comparison of identified clusters with adult mouse brain data sets.

While this manuscript reports a resource that has the potential for being extremely useful for the community, the inclusion of only two developmental time points (E14.5 and P0) remains a significant limitation of this study. While the authors have explained their choice, and these two time points certainly make sense to include, the lack of other time points severely limits the utility of this resource. This weakness dampens support for an otherwise strong manuscript.

We thank the reviewers for their insightful feedback. We have revised our manuscript and addressed each of the points below:

Reviewer #3 (Remarks to the Author):

The manuscript by Loo et al., has improved upon revision. The Drop-seq data are high quality, the clustering/sub-clustering methods for both ages are sound, and the cluster interpretations are well reasoned (with a few possible exceptions below). Most of the tissue validation is still based on the expression of single genes, rather than proving co-expression patterns, but the authors do provide staining to validate co-expression patterns during the differentiation and migration of Cajal-Retzius neurons from the cortical hem. Like many recent single cell papers, the text necessarily includes a tour of cluster and defense of interpretations/disease implications, but I think there is potential to also connect more to recent themes cortical development.

I still have several comments that the authors can likely address:

- 1) Are the authors suggesting that 5-E and 13-E represent area-specific subsets of *Fezf2* neurons? If so, this should be made more explicit and analyzed more deeply. Nowakowski et al., 2017 recently showed that in human, area-specific excitatory neurons emerge early in cortical development prior to sensory experience. The author should reference this work and address whether the same markers delineate these subclusters. In addition, both the Tasic paper in adult mouse, and the Nowakowski paper in developing human cortex sampled extreme poles of the cortex, leaving open the question of whether there are discrete types or whether the excitatory neurons vary along a gradient. In this study, was the whole rostrocaudal extent of the brain sampled? And could 5-E and 13-E neurons be better represented along a gradient? Also, why is there only a single *Fezf2* cluster at the P0 timepoint – in the Tasic paper, it looks like areal differences may be greater in adult? Does this cluster contain neurons that appear more rostral or caudal? Finally, do the upper layer excitatory neurons also appear to be area-specific. To summarize, if the authors are advancing the possibility of area-specific excitatory neurons, which has recently emerged in the field, there needs to be more analysis and comparison with other findings.

Yes, both 5-E and 13-E represent *Fezf2*+ populations, however their other markers seem to be indicative of area specificity. As we describe, *Crym* and *Mc4r* expression delineate caudal and rostral cells, respectively. To better illustrate this, we have now included ISH data from Allen Brain Atlas at both E15.5 and E18.5 (**Supplementary Fig. 14**) to show a caudal>rostral gradient for these two marker genes.

- 2) The 3-E cluster is called Layer 5, but expresses high levels of *Satb2*, *Pou3f1*, and *Cux2*, no *Fezf2*, and lower levels of *Bcl11b* and *Tbr1*. These might be immature upper layer neurons or a mature intratelencephalic cluster.

As the authors note later in the text, upper layer neurogenesis does start around E14.5. In addition, neuronal maturation markers are lower in this cluster, so my guess is that these are indeed immature upper layer neurons.

We agree that these cells are likely in transition and are destined to become upper-layer neurons. Since these cells express markers of both upper and lower-layer neurons, we maintained the “Layer V–VI” label in Fig. 1–2, however a description of this duality is mentioned and cited in the Results section:

We also observed novel correlations that, when combined with the underlying expression patterns, are suggestive of early fate specification. For example, Cluster 3-E expressed an upper-layer CPN marker (*Satb2*), a lower-layer marker (*Bcl11b*), a migratory marker *Tiam2*, and *Pou3f1*, a transcription factor that is expressed in Layer II–III neurons during their migration and differentiation (**Supplementary Fig. 4, 6, and 10**) {Frantz, 1994 #11; Molyneaux, 2007 #1822}. This cluster was most similar to an upper-layer CPN at P0 (Layer II–IV; Cluster 1-P). These cells may therefore be destined to become upper-layer CPN, some of which are known to be born around E14.5 (Ref. {Telley, 2016 #13}).

3) Cluster 8P is surprising in that it contains markers of excitatory neurogenesis (Eomes, Neurod6) along with Calb2. I believe that Calb2 is not usually observed in excitatory neurons, except for Cajal-Retzius cells, but this is interpreted as an SVZ cluster and captured at a stage when we might not expect new Cajal-Retzius neurons to be generated. The in situ data from Allen provided in the supplement shows Calb2 most strongly staining marginal zone. If the authors could provide co-expression data from SVZ of Neurod6 (or other excitatory lineage markers) and Calb2, this would be helpful in confirming this unexpected population. Alternatively, 8P could represent doublets between Cajal-Retzius and SVZ progenitors (2-P), or there is another interpretation I am unaware of.

We performed immunofluorescence staining in P0 *Neurod6:CRE* mice to confirm the existence and localization of this population of cells. We used these mice due to NEUROD6 antibody limitations, however the expression of CRE faithfully recapitulates endogenous *Neurod6* promoter activity³². We observed a migratory stream of NEUROD6/CRE⁺ CALB2⁺ cells in the corpus callosum as well as some similarly labeled cells in the cortical plate (shown in new **Supplementary Fig. 15**). These cells have similar localization and expression patterns to the Rostral Migratory Stream, a population of cells that migrate to the cortex and olfactory bulb postnatally³³⁻³⁴.

These data are now described in the modified Results section, and we believe that the validation of co-labeling demonstrates that these cells are not doublets captured during sample preparation.

4) Why are 10P and 13P considered astrocyte IPCs when they do not express cell cycle markers? Would it not make more sense to consider these astrocytes? Also, the *Aqp4* in situ in S12 seems to not show expression.

These astrocytic cells still express radial glial markers *Hes1* and *Hes5* (Sup. Fig. 4), so despite expressing *Aqp4* and *Aldh1l1*, and having low expression of cell cycle markers, we believe they still require a qualifying label indicating they are not mature astrocytes. We have therefore modified their name throughout the manuscript to “Astrocytes (immature)”.

We have also modified Sup. Fig 12 to better demonstrate *Aqp4* expression in the VZ/SVZ.

5) I am surprised that the SST/Lhx6 interneurons don't translate well across species or age. Nowakowski et al., reported both CGE and MGE-derived inhibitory neurons in cortex, and markers like SST, Lhx6, and Nkx2-2 are likely conserved. Have other genes changed to an extent to limit homology, or is this a limitation of the correlation method? (The visualization of correlations between different datasets is very nice)

Due to point #6 below, this analysis changed somewhat, and the conclusion that we draw now is that both SST and VIP interneurons of the P0 cortex correlate with human interneurons of CGE/MGE origin (Nowakowski et al.). We have updated the Results section accordingly. However, the lack of strong correlation among all interneuron populations across developmental time (within mouse) remains, therefore, the description in the Discussion about possible reasons for this disparity has been maintained.

6) The timepoints comparing E14.5 mouse to human are not well aligned and must be changed. E14.5 mouse corresponds to PCW12.5 to PCW14.5, when deep layer neurogenesis starts to wind down in human. PCW7-9 is much too early. This tool provides one method for conversion: <http://translatingtime.org/translate>

The exact alignment of mouse to human brain developmental age seems to depend somewhat on the reference. According to Carnegie staging, which is more holistic rather than brain-specific (https://embryology.med.unsw.edu.au/embryology/index.php/Carnegie_Stage_Comparison), E14.5 corresponds to post-conception day 52 in human. Similarly, according to Otis et al 1954 (PMID 13207763), E14.5 corresponds to post-conception day 51.5 (7.5pcw) in human. Using <http://translatingtime.org> as the reviewer suggests, we see that sensorimotor and other events in the cortex at mouse E14.5 correspond to post-conception day 67 in human (9.6pcw). Further, if you start with human PC day 63 (9pcw), it translates to mouse E14, whereas the reviewer suggestions of 87.5 (12.5pcw) or 101.5 (14.5pcw) translate to E15-18, which is seemingly too late. Nonetheless, if

neurogenesis is the focus, then we do see that E14.5 may correspond to as late as PC day 87 in the human (12.5pcw). Note that the Nowakowski et al dataset does not include any cells from donors >9 and <11 pcw, or from >11.5 and <13 pcw.

As such, we have repeated our correlation analysis now ranging from 7-11.5pcw to capture a more complete range of human developmental timing and marker gene expression. We have updated Supplementary Figure 16 to reflect this change.

7) vRG probably is meant to refer to ventricular radial glia, rather than radial glia in ventral telencephalon

The text has been edited to reflect this change

8) In figure 5, does layer-specific mean excitatory neuron?

We were careful to use the term “layer-specific” throughout the manuscript when referring to cells comprising each of the cortical layers due to the fact that our “Layer I” cluster contains multiple subpopulations. Though many are mature (excitatory) Cajal-Retzius cells, this cluster also contains their precursors that we showed are in transition from the cortical hem. These precursors, however, don’t seem to express VGlut2 (*Slc17a6*) yet, so we were cautious to not refer to them as “excitatory neurons”. We feel the term “layer-specific” is more appropriate here as it unambiguously refers to all cells labeled as Layers I–VI regardless of their state of maturation.

9) I would probably combine figure 4 and 5 as Figure 4 is an example that is generalized (and included) in Figure 5

A combined figure might make sense conceptually, however we feel it is too busy and therefore confusing to readers, so we opted to keep them separated.

10) The web tool is great – the dataset and interpretations will be more valuable for distributing the data in this way. Can you also let users pick a cluster and get all the cluster markers? Also, can you allow a user to input a list of genes and get table of values? Alternatively, the supplement is fine for this purpose.

For now, we prefer readers to go to the Supplemental data and/or code we have provided for this purpose, however, this functionality may be incorporated into a future release of the web tool.

11) I appreciate the supplementary table linking each interpretation to genes and literature searches or in situs.

We thank the reviewer for this comment.

Reviewer #4 (Remarks to the Author):

This manuscript by Loo et al. makes use of single cell RNA-seq to generate a transcriptomic description of cell types in the developing mouse cerebral cortex at E14.5 and P0. Validation of identified clusters is done through ISH and IHC.

Overall, the authors have substantially revised the manuscript in response to the prior review comments. Significant improvements include: the development of a web-based interface for data visualization, improved consideration of other studies using similar methodologies and comparison of identified clusters with adult mouse brain data sets.

While this manuscript reports a resource that has the potential for being extremely useful for the community, the inclusion of only two developmental time points (E14.5 and P0) remains a significant limitation of this study. While the authors have explained their choice, and these two time points certainly make sense to include, the lack of other time points severely limits the utility of this resource. This weakness dampens support for an otherwise strong manuscript.

We thank the reviewer for this feedback.

REVIEWERS' COMMENTS:

Reviewer #3 (Remarks to the Author):

The authors have put together a high quality manuscript and responded to my main concerns. In particular, the expression of Calb2 in Neurod6 lineage cells is a nice touch to validate co-expression patterns in an unexpected cell type. This paper will be a valuable resource and includes interesting discussion of developmental gradients and coherence of disease related markers. I would still remove the word "comprehensive" from the first sentence of the discussion as only a few time points are measured and it is not clear that rare spatially restricted cell types would be captured.

Response to reviewers

Reviewer #3 (Remarks to the Author):

The authors have put together a high quality manuscript and responded to my main concerns. In particular, the expression of Calb2 in Neurod6 lineage cells is a nice touch to validate co-expression patterns in an unexpected cell type. This paper will be a valuable resource and includes interesting discussion of developmental gradients and coherence of disease related markers. I would still remove the word “comprehensive” from the first sentence of the discussion as only a few time points are measured and it is not clear that rare spatially restricted cell types would be captured.

Thank you for your comments. We have removed the word “comprehensive” from the Discussion as requested.